EMBO
Molecular Medicine

# Uncovering biomarkers for chronic toxoplasmosis detection highlights alternative pathways shaping parasite dormancy

Marie G Robert[1,2,5], Christopher Swale [ID][1,5], Belen Pachano[1], Léa Dépéry[1], Valeria Bellini[1], Céline Dard[1], Dominique Cannella[1], Charlotte Corrao[1], Lucid Belmudes[3], Yohann Couté [ID][3], Alexandre Bougdour [ID][1], Hervé Pelloux[1,2], Emmanuelle Chapey [ID][4], Martine Wallon [ID][4], Marie-Pierre Brenier-Pinchart [ID][1,2,6 ✉] & Mohamed-Ali Hakimi [ID][1,6 ✉]

## Abstract

***Toxoplasma gondii***, a neurotropic protozoan, causes toxoplasmosis, a prevalent zoonotic and food-borne infection, posing significant risks to immunocompromised individuals and congenital cases. The chronic phase, characterized by dormant, cyst-forming bradyzoites, is central to disease progression but is poorly understood due to the lack of serological tests to detect bradyzoite-specific antigens. This study identifies the bradyzoite serological marker (BSM) and cyst-associated BCLA as effective biomarkers for chronic toxoplasmosis. These markers showed high sensitivity and specificity in detecting cyst-bearing mice and had a positivity rate of 30% in humans with prior immunity. Bradyzoite serology helps to discriminate between recent and past infections, with BCLA improving the accuracy of the diagnosis of congenital infections. Mechanistic analyses show that the chromatin modifiers MORC and HDAC3 epistatically regulate BFD1, a key bradyzoite regulator. While BFD1 controls the expression of bradyzoite genes such as BCLA, a specific subset, including BSM, is regulated independently of BFD1. This multilayered regulation complicates the understanding of parasite persistence in humans, but offers promise for improved serologic diagnosis during pregnancy, but also in individuals with mental illness.

**Keywords** Bradyzoite; Epigenetics; Serology; *Toxoplasma gondii*; Transcription
**Subject Categories** Biomarkers; Immunology; Microbiology, Virology & Host Pathogen Interaction

## Introduction

*Toxoplasma gondii* spreads successfully worldwide by reproducing sexually in cats and asexually in all warm-blooded animals, including humans. Infection occurs through the fecal-oral route or carnivorism. In the intestinal epithelium of cats, ingested bradyzoites transform and initiate the sexual cycle, resulting in the production of oocysts. These oocysts are shed in the cat's feces and sporulate in an aerobic environment. When ingested by humans, the oocysts release sporozoites, which develop into tachyzoites that spread the infection to organs such as the brain. The immune response then prompts tachyzoites to become dormant bradyzoites in cysts within muscles, heart, brain, and eyes (Dubey et al, 1998). In the central nervous system (CNS), various resident cell types may interact with tachyzoites, but only neurons support cyst development (Cabral et al, 2016; Melzer et al, 2010). The cycle completes when a cat consumes a cyst-bearing animal. The infection commonly found in livestock poses serious risks, especially for small ruminants (Stelzer et al, 2019), leading to major economic losses and acting as a source for transmitting the infection to humans (Kijlstra and Jongert, 2009). In humans, seroprevalence is estimated to exceed 30% worldwide, with regional variations influenced by dietary habits, hygiene practices, and local environmental conditions (Bigna et al, 2020; Pappas et al, 2009).

Toxoplasmosis is generally mild or infraclinical in immunocompetent individuals but primary maternal infection during pregnancy is a significant cause of congenital infection, posing a potentially lethal risk to the fetus. Toxoplasmosis can become notably severe and even life-threatening in immunocompromised individuals, often due to the reactivation of cysts and differentiation of bradyzoites into tachyzoites. T-cell lymphopenia from HIV/AIDS, immunosuppressive therapy, or hematological malignancies sharply increases the risk of *T. gondii* encephalitis (TE), a severe neuroinflammatory disease still common in immunosuppressed patients, including those with HIV (Ondounda et al, 2016).

[1]Team Host-Pathogen Interactions and Immunity to Infection, Institute for Advanced Biosciences, INSERM U1209, CNRS UMR5309, Grenoble Alpes University, Grenoble, France. [2]Laboratory of Parasitology and Mycology, Grenoble Alpes University Hospital, CS10217, 38043 Grenoble, France. [3]Grenoble Alpes University, INSERM, CEA, UA13 BGE, CNRS, CEA, FR2048, 38000 Grenoble, France. [4]Hospices Civils de Lyon, Institut des Agents Infectieux, Lyon, France. [5]These authors contributed equally: Marie G Robert, Christopher Swale. [6]These authors contributed equally: Marie-Pierre Brenier-Pinchart, Mohamed-Ali Hakimi. ✉E-mail: MPPinchart@chu-grenoble.fr; mohamed-ali.hakimi@inserm.fr

Interestingly, the risk of reactivation from cysts is not limited to immunocompromised individuals, as both congenital and acquired toxoplasmosis can lead to ocular complications long after the initial infection (Delair et al, 2008). As a result, toxoplasmosis stands as a primary reason for posterior uveitis worldwide (Furtado et al, 2013). Critically, without a drug targeting the encysted bradyzoite stage, this parasite cannot be cleared from the brain and retina.

Toxoplasmosis diagnosis relies heavily on detecting antibodies targeting *T. gondii* or parasite DNA (Dard et al, 2016; Robert et al, 2021). Serological methods focus on detecting antibodies against tachyzoite antigens, which limits their overall effectiveness. Current serologic tests mainly use native tachyzoite antigens, but recombinant antigens are studied to improve accuracy and standardization (Ybañez et al, 2020). Tachyzoite-based serology relies on the kinetics of different antibody isotypes, mainly IgG and IgM, to assess infection status and risk in cases like maternal–fetal transmission or transplantation but struggles with precisely dating infections—a critical need in certain scenarios (Dard et al, 2016). Bradyzoites within cysts play a central role in host-to-host transmission and reactivation in chronic infection, yet they are largely ignored in serologic testing. In the absence of a straightforward, noninvasive method to confirm carriage of the chronic form, diagnosis paradoxically depends on detecting IgG antibodies directed against tachyzoite antigens, which are usually markers of acute infection. The belief that infections always result in lifelong cysts is being challenged (Rougier et al, 2017), as cyst formation and persistence in humans remain poorly understood. Once thought to be static, cysts are dynamic structures; bradyzoites within them periodically replicate, expanding cyst size (Watts et al, 2015). Over time, cysts rupture, releasing bradyzoites that are either cleared by the immune system or form new daughter cysts (Frenkel and Escajadillo, 1987). In addition, chronic *T. gondii* infection in the brain may be associated with neuropsychiatric disorders, such as schizophrenia and cognitive changes, although there are conflicting data on this issue (Johnson and Koshy, 2020; Sutterland et al, 2015). In rodents, the natural hosts of *T. gondii*, the parasite has been shown to influence the progression of neurodegenerative disorders (Cabral et al, 2017) and induce significant behavioral changes (Boillat et al, 2020; Dupont et al, 2021).

Detecting cyst carriage remains a challenge. While Zhang et al (1995) suggested that anti-*Toxoplasma* antibodies primarily target tachyzoites (Zhang et al, 1995), subsequent studies have offered a more nuanced perspective. The matrix antigen MAG1 (*TGME49_270240*), located in the cyst wall, has been shown to elicit an early humoral immune response upon human infection (Di Cristina et al, 2004). Further studies have correlated the presence of anti-MAG1 antibodies with chronic toxoplasmosis (Xiao et al, 2013, 2016, 2021). However, MAG1 is not exclusive to bradyzoites and cysts. It is also expressed in tachyzoites, albeit at lower levels (Ferguson and Parmley, 2002). Genome-wide and proteome-wide analyses have further confirmed the presence of both MAG1 mRNA and protein in tachyzoites (source: ToxoDB.org). While MAG1 may be more detectable serologically in chronic toxoplasmosis, its broad expression undermines its reliability as a specific marker for chronic infection.

In contrast, BCLA/MAG2, which localizes similarly to MAG1 at the cyst periphery, is exclusively expressed by bradyzoites. As such, BCLA has been identified as a promising biomarker for cyst detection (Dard et al, 2021) and a breakthrough in understanding chronic infection (Tu et al, 2020). BCLA has consistently demonstrated reliability as a biomarker for cyst carriage in mice with controlled infections. In humans, BCLA ELISA showed strong reactivity in individuals with past immunity, especially in those with cyst-associated conditions like ocular toxoplasmosis. However, variability in titers and occasional unexpected results in seropositive and seronegative patients add interpretive complexity, as human comparisons rely solely on tachyzoite markers, unlike in mouse models (Dard et al, 2021).

In this study, we addressed the limitations of BCLA by expanding the diagnostic toolkit for chronic toxoplasmosis. We identified the Bradyzoite Serological Marker (BSM)—also known as DnaK-TPR (Ueno et al, 2011; Yang et al, 2017)—by upscaling the production of bradyzoite antigens through epigenetic reprogramming of tachyzoites in vitro (Farhat et al, 2020). We then evaluated the diagnostic accuracy of BCLA and BSM markers in mice and human sera and compared their performance with conventional serological methods. In exploring the control mechanisms behind BSM expression, we uncovered that MORC/HDAC3 exerts epistatic control over BFD1, a regulator of bradyzoite development (Waldman et al, 2020). This BFD1-mediated pathway controls the expression of a subset of bradyzoite-specific transcripts, notably BAG1 and BCLA. However, BSM falls under a separate sub-transcriptome that, while still reliant on MORC/HDAC3, does not involve BFD1. This finding underscores the diversity of molecular pathways and their regulatory checkpoints that control cyst development and persistence in tissues, leading to a range of immune responses in hosts and complicating the diagnosis of chronic toxoplasmosis in humans.

# Results

## Identification of immunogenic markers in bradyzoite proteins enriched from MORC knockdown extracts

In tachyzoites, MORC partners with histone deacetylase HDAC3 to restrict chromatin access to transcription machinery at genes specific to the sexual and chronic stages (Antunes et al, 2024; Farhat et al, 2020). Depletion of MORC results in significant transcriptional changes, leading to the expression of a large repertoire of bradyzoite-specific genes, encompassing up to 50% of the chronic sub-transcriptome (Farhat et al, 2020). Inhibition of HDAC3 by the HDAC inhibitor FR235222 phenocopies the depletion of MORC, inducing transcriptionally a transition from tachyzoites to bradyzoites (Bougdour et al, 2009; Farhat et al, 2020). Producing large amounts of immunogenic bradyzoite proteins has been a challenge, limiting discovery. This bottleneck can now be overcome by inducing bradyzoite development in vitro without stressors (e.g., alkaline pH), through overexpressing (e.g., BFD1) or suppressing (e.g., MORC/HDAC3) key genetic switches (Waldman et al, 2020; Farhat et al, 2020). Proteomic analysis after MORC knockdown (KD) revealed an increase in proteins matching bradyzoite gene expression, including key markers BAG1 and BCLA/MAG2, hallmarks of bradyzoites and latent tissue cysts (Farhat et al, 2020; Dard et al, 2021). We hypothesized that reducing MORC levels could be an effective strategy to discover new immunogenic bradyzoite markers. To test this, we analyzed the immunoreactivity of sera from chronically infected, BCLA-positive

mice (Dard et al, 2021) across fractions from five conventional chromatographic steps that separate and concentrate MORC-depleted protein extracts (Fig. EV1A). Despite strong immune responses, only MORC depletion revealed a consistent ~80 kDa protein band across different parasite genetic backgrounds in sera from chronically infected mice (Fig. 1A). After 30% w/v ammonium sulfate precipitation, the MORC-depleted protein was further purified using spin filtration with a 30 kDa cutoff and gel filtration chromatography (Fig. EV1A,B). Silver stain analysis of the final chromatographic step showed a nearly homogeneously purified polypeptide(s) with a size of 80 kDa (Fig. 1B), which elicited a strong and specific reaction with sera from chronically infected mice (Fig. 1C). Mass spectrometry (MS)-based proteomic analysis confidently identified the proteins TGME49_216140 and TGME49_202020, based on their abundance, predicted molecular weight of 62 and 83 kDa, respectively, and their restricted expression during the bradyzoite stage (Fig. EV1C). Notably, both proteins are upregulated in MORC-depleted cells (Farhat et al, 2020).

## TGME49_202020, a bradyzoite-expressed protein with immunogenic properties

To better document the silencing of their expression in tachyzoites by MORC, we endogenously Flag-tagged *TGME49_216140* and *TGME49_202020* in the RH MORC-mAID-HA KD lineage. As anticipated, treatment of these edited parasites with IAA led to the near elimination of MORC, followed by the accumulation of the FLAG-tagged version of the TGME49_216140 and TGME49_202020 chimeric proteins (Fig. EV1D). Both proteins were then immunopurified from MORC-depleted extracts using anti-Flag affinity chromatography, resolved by SDS-PAGE, and analyzed by Western blot to assess their specific humoral response to infected mouse sera. TGME49_216140 is clearly nonimmunogenic and migrates below the 60 kDa marker (Fig. EV1E). In contrast, the immunopurified TGME49_202020 migrates above the 60 kDa marker, aligning with the electrophoretic profile of the enriched immunogenic protein (Figs. 1B and EV1F). Notably, this purified protein is specifically recognized by sera from mice chronically infected with type II strains that carry brain cysts (ME49 and 76 K), but not by sera from NMRI mice infected with CTG, a type III strain characterized by a low cyst load at the limit of detection (Fig. 1D) (Dard et al, 2021). Given its restricted expression to bradyzoites (Fig. EV1C) and its reactivity with sera from mice with brain cysts, we henceforth refer to TGME49_202020 to as the Bradyzoite Serological Marker (BSM).

This protein has also been independently identified as DnaK-TPR, referring to its heat shock protein (DnaK) and tetratricopeptide repeat (TPR) domains (Ueno et al, 2011) (Fig. 1E,F). The structure of BSM, as predicted by AlphaFold2 and sourced from the EBI/AlphaFold repository, features the previously mentioned DnaK and TPR domains intricately fold against each other (Fig. 1E,F), suggesting potential allosteric regulation of client protein binding by the TPR domain. The closest experimentally determined structural homolog to the DnaK domain is a bacterial Hsp70 from *E. coli* K12, as identified by the Foldseek structural search engine (van Kempen et al, 2022) with a Tm score of 0.71 (a Tm score above 0.5 indicates significant homology), despite only 10.5% sequence homology. The TPR domain, on the other hand, partially

aligns with segments of human FKBP38 with a Tm score of 0.55 and 7.3% sequence identity. Overall, these structural predictions confidently suggest that this DnaK-TPR protein functions as an ATP-dependent chaperone, potentially involved in bradyzoite-specific protein folding pathways.

## BSM accumulates in the cytoplasm of bradyzoites converted in vitro or found in brain cysts from mice

To further investigate the subcellular location of BSM, we have developed polyclonal antibodies against the protein. Following MORC depletion or HDAC3 inhibition, BSM was detected exclusively in the cytoplasm of in vitro-induced bradyzoites, within vacuoles that accumulate Dolichos biflorus agglutinin (DBA) lectin on their surface (Fig. 2A), as a consequence of the induction of CST1-SRS44 - a cyst wall protein that is responsible for the reactivity of DBA (Tomita et al, 2013). No BSM was detected in the 76 K strain treated with FR235222, which was genetically modified to lack BSM (Δ*bsm*), confirming the specificity of our in-house antibody (Fig. 2B). Meanwhile, the expression of BCLA in the vacuolar space and at the parasite vacuolar membrane (PVM) remained unaffected by the absence of BSM when induced by FR235222 (Fig. 2B). Considering that full maturation of bradyzoites and complete cyst biogenesis of *T. gondii* cannot be achieved in vitro (Milligan-Myhre et al, 2016), we also examined the presence of BSM in histological brain sections from mice chronically infected with a cystogenic strain and confirmed the cytoplasmic localization of BSM in fully differentiated bradyzoites (Fig. 2C; Movie EV1).

## MORC/HDAC3 dual regulation: epistatic control over BFD1 and independent regulation of bradyzoite genes

BSM and BCLA mRNA were captured both in bulk and at the single-cell level from in vitro-induced and in vivo-harvested cysts/ bradyzoites (Source: ToxoDB). Single-cell transcriptome analysis of tachyzoite and bradyzoite (Xue et al, 2020) revealed that BSM and BCLA predominantly segregate within the P1 population, identified as the bradyzoite-specific cluster (Fig. 2D). Interestingly, BSM is also significantly expressed in a distinct sub-population of zoites, labeled P6 (Fig. 2D), which may represent a previously unrecognized intermediate state in the asexual life cycle (Xue et al, 2020). This difference cannot be attributed to MORC and HDAC3, as both are involved in regulating the expression of these markers without distinction (Fig. 2A,B). We then explored upstream transcription factors that might be involved. In *Toxoplasma*, the transcription factor BFD1 has been reported as both necessary and sufficient to recapitulate the transcriptional changes seen in naturally derived bradyzoites when conditionally overexpressed (Waldman et al, 2020). Despite being transcribed in tachyzoites, BFD1 protein is not detected, suggesting translational control, later shown to be mediated by BFD2/ROCY1 (Licon et al, 2023; Sokol-Borrelli et al, 2023; Wang et al, 2024).

We first looked at a potential link between MORC/HDAC3 and BFD1. Analysis of previous proteomic datasets revealed that the BFD1 protein accumulates after inhibition of HDAC3 by FR235222. We confirm the induction and nuclear localization of BFD1 protein not only by the chemical action of FR235222, but also by the depletion of MORC (Fig. 2E,F). BFD1 expression starts

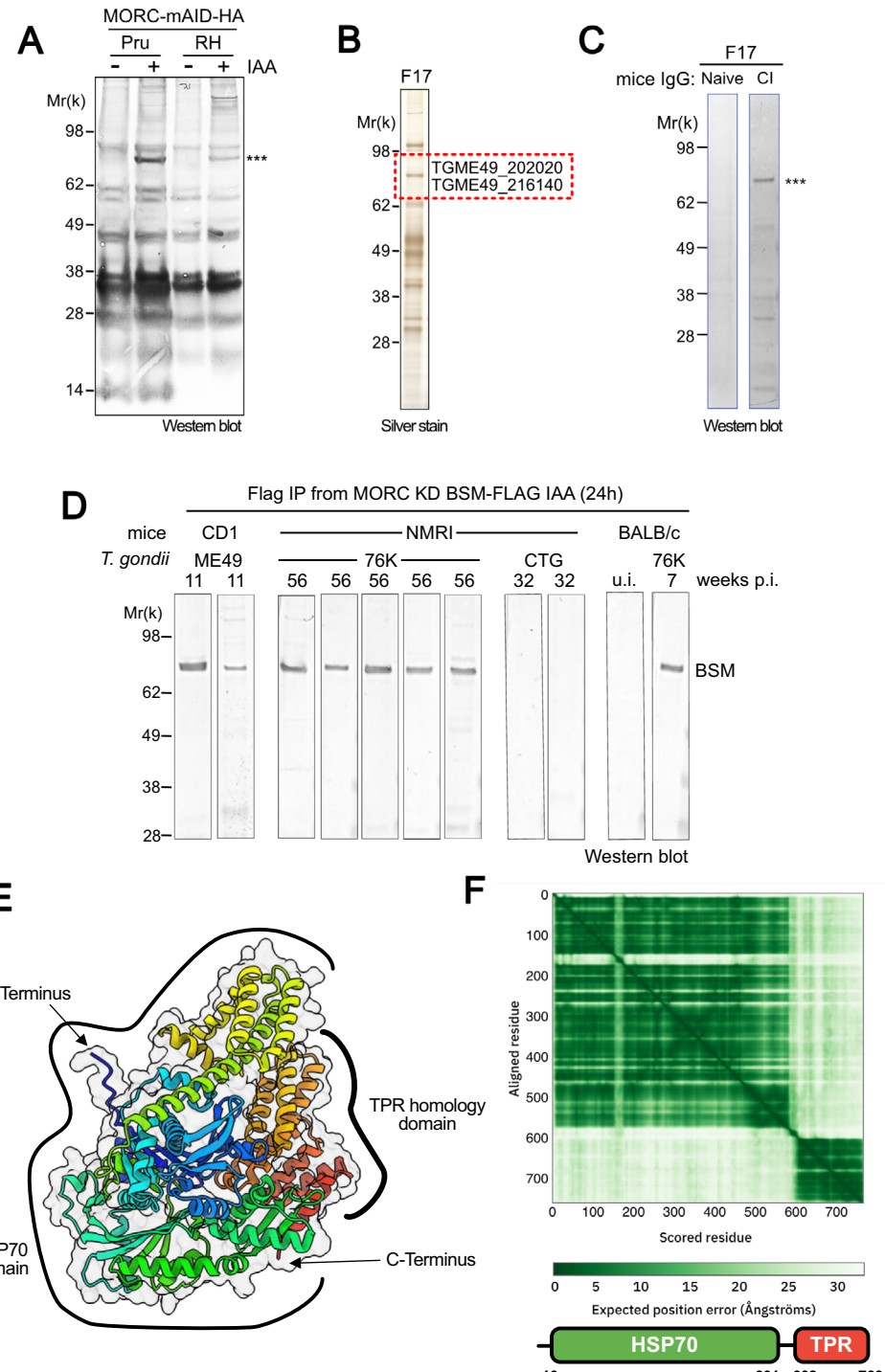

**Figure 1. Discovery of bradyzoite-specific immunogenic proteins in MORC knockdown extracts.**

(A) Western blot membranes from IAA-treated and untreated extracts were probed with antibodies from mice chronically infected with cystogenic *T. gondii* strains. Results were consistent across replicates, and a representative blot is shown. (B) Silver staining of fraction 17 (F17) from S200 gel filtration chromatography (full column fractions shown in Fig. EV1B). (C) Western blot of F17 probed with sera from naive mice and mice chronically infected with *T. gondii*. (D) BSM was immunopurified using anti-Flag antibodies, and Western blots of the purified protein were probed with sera from chronically infected mice bearing *T. gondii* cysts (CD1 infected with ME49, NMRI, and BALB/c infected with 76 K) or from NMRI mice infected with the cyst-free strain CTG. (E) Alphafold model of the BSM protein (generated via the EBI/Alphafold server), displayed as a cartoon and color-coded from the N-terminal (blue) to the C-terminal (red). (F) Alphafold position error heatmap illustrating the predicted subdomains of BSM. Source data are available online for this figure.

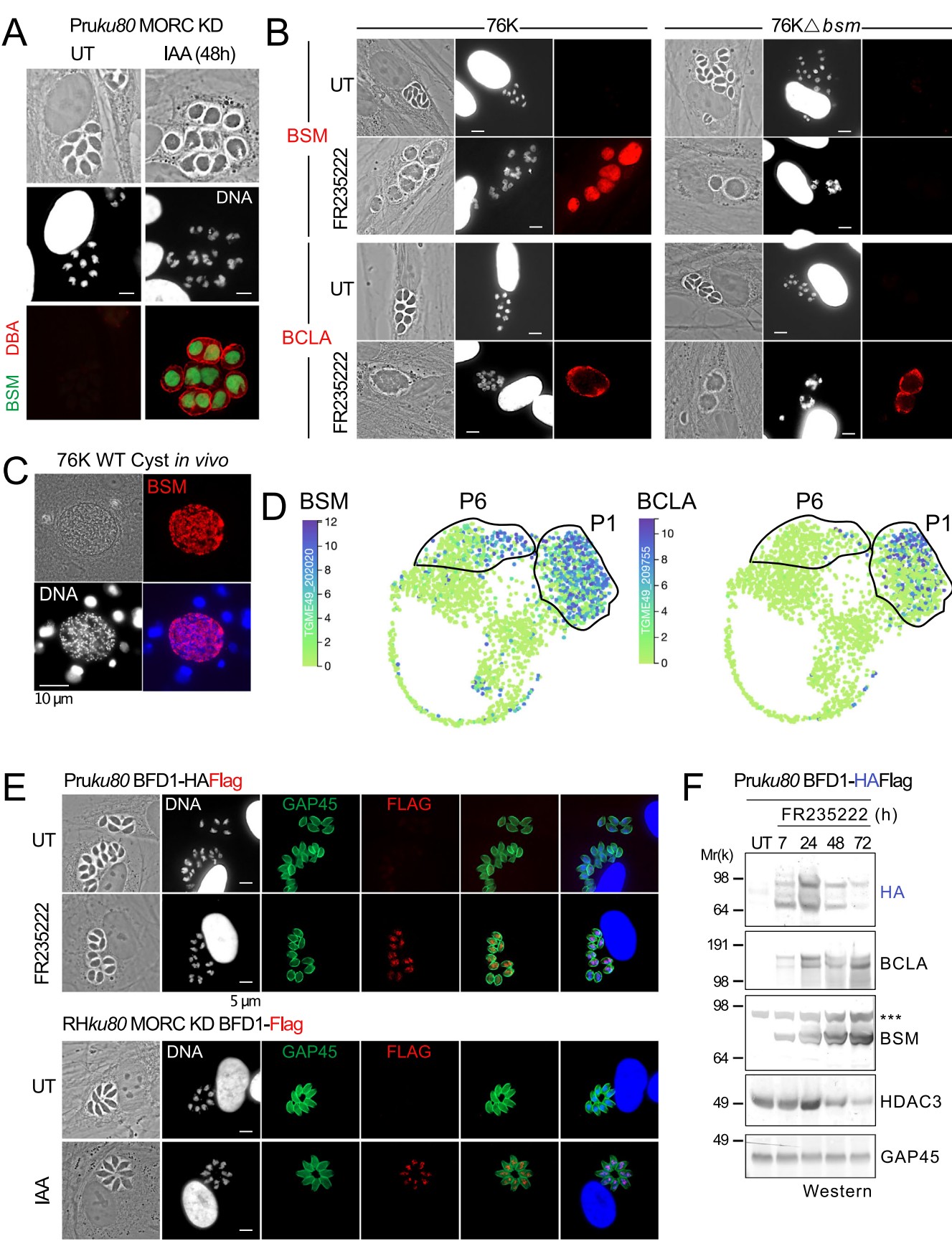

**Figure 2. MORC and HDAC3 regulate the expression of bradyzoite markers BSM and BCLA, along with the transcription factor BFD1.**

(A) Expression of BSM (green) following MORC KD in the type II strain. The cells were co-stained with DBA lectin (red) and DNA-specific Hoechst dye (white). Scale bar = 5 μm. (B) The expression levels of BSM and BCLA in 76 K WT or Δ*bsm* strains were assessed by IFA following HDAC3 inhibition with FR235222. Scale bar = 5 μm. (C) BSM (in red) was identified by IFA on a cyst in a brain histological section from a mouse chronically infected with the type II strain (76 K). The cells were co-stained with DNA-specific Hoechst dye (white or blue). (D) UMAP projection of Pru tachyzoite and bradyzoite scRNA experiment (Xue et al, 2020). P1 population corresponds to the bradyzoite-specific cluster. (E) Expression of BFD1 (red) following HDAC3 inhibition by FR235222 (upper panel) or MORC KD (lower panel). The cells were co-stained with GAP45 (green) and DNA-specific Hoechst dye (white or blue). (F) Time-course analysis of the expression levels of BFD1 following inhibition of HDAC3. The samples were taken at the indicated time periods after the addition of FR235222 and were probed with antibodies against HA and HDAC3 as well as bradyzoite proteins BCLA and BSM. The same experiment was repeated three times, and a representative blot is shown. Source data are available online for this figure.

as early as 7 h after FR235222 addition, peaks at 24 h, and then declines at later time points (Fig. 2F), likely due to the simultaneous development of sexual stages (Antunes et al, 2024; Farhat et al, 2020). Inactivation of MORC/HDAC3 leads to widespread transcriptional changes, yet it does not affect BFD1 or BFD2/ROCY1 mRNA levels (Farhat et al, 2020), which remain stable throughout the asexual cycle. This indicates that the MORC/HDAC3 complex indirectly exerts epistatic control over BFD1 by repressing an as-yet-unidentified factor that regulates the translation or degradation rates of BFD1. Notably, MORC/HDAC3 does not influence BFD2/ROCY1, which might have been an expected target. To further examine this relationship and its impact on bradyzoite development, we investigate how the MORC/HDAC3 signaling pathway interacts with BFD1, using BCLA and BSM as markers for bradyzoite development.

To assess whether BFD1 is required to regulate BCLA and BSM expression upstream of MORC, we replaced the Myb DNA-binding domain of BFD1 with a DHFR cassette in a MORC KD background to generate a MORC KD/*BFD1* KO strain. In the presence of functional BFD1, MORC depletion successfully induces BSM, BCLA, and DBA/CST1 (Fig. 3A). However, disruption of the DNA-binding capacity of BFD1 specifically blocks MORC-driven induction of BCLA and DBA/CST1, while BSM expression remains notably unaffected (Fig. 3A). This suggests that BSM expression is regulated by MORC/HDAC3 through a BFD1-independent mechanism. To validate this, we took the opportunity of a strain expressing a regulatable version of the BFD1 protein (ΔBFD1/DD-BFD1-Ty (Waldman et al, 2020)). In this background, BFD1 is continuously degraded post-translationally unless Shield-1 is added to the culture media to stabilize the degradation domain (DD). Treatment with Shield-1 leads, as expected, to robust nuclear accumulation of DD-BFD1-Ty (Fig. 3B), effectively activating several bradyzoite markers, including DBA/CST1, BAG1, SRS35A/SRS4, and BCLA. However, BSM expression remains undetectable 72 h after Shield-1 addition (Fig. 3C). This clearly demonstrates that BFD1 does not act downstream of MORC/HDAC3 in the transcriptional regulation of BSM.

To explore the extent to which MORC/HDAC3 and BFD1 intersect to control bradyzoite development beyond the narrow scope revealed by BSM and other co-staining markers, we conducted sequencing of bulk mRNA in both MORC KD and MORC KD/*BFD1* KO strains (Dataset EV1). Principal component analysis showed that addition of IAA was responsible for most of the variance between samples (Fig. 4A). Using DESeq2 to analyze the RNA-seq data, we identified 1788 genes (FC ≥ 2 and *P* value < 0.05) that were specifically silenced by MORC (Fig. EV2A), aligning with our previous study (Farhat et al, 2020). While absent

in the tachyzoite stage, these genes are predominantly expressed in sexual stages and, notably, only a few are restricted in their expression to the bradyzoite stage. Among the genes present in chronic stages, we identified two clusters: those dependent on BFD1 (e.g., BCLA) and those independent of BFD1 (e.g., BSM) (Fig. 4B). Using DESeq2 to analyze mRNA levels between IAA-treated MORC KD and IAA-treated MORC KD/*BFD1* KO, we found 83 genes that were upregulated following MORC depletion only in the presence of functional BFD1, including the canonical bradyzoite markers BAG1, ENO1, and BCLA (Fig. 4C).

MORC/HDAC3 has been established as a repressor at strategic checkpoints along the life cycle continuum (Bougdour et al, 2009; Farhat et al, 2020; Antunes et al, 2024). MORC degradation leads to enhanced chromatin accessibility, as shown by ATAC-seq (Assay for Transposase-Accessible Chromatin using sequencing) (Pachano et al, 2025). In the context of genes exclusively expressed in chronic stages and regulated by BFD1, MORC was not detected at their promoters by ChIP-seq (Fig. 4D,E). However, its depletion results in increased chromatin accessibility and subsequent gene transcription (Fig. 4D,E). Here, MORC indirectly stimulates the expression of the BFD1 protein (Fig. 2E), which in turn facilitates the establishment of an open chromatin state near these genes. At BFD1-independent genes expressed exclusively in chronic stages, a similar pattern of chromatin opening occurs, but this is expected to be driven by a different, as-yet-unidentified, transcription factor (Fig. 4F,G). For genes expressed in both merozoite and bradyzoite, such as BRP1 and ROP26, we have reported local co-occupancy of MORC and HDAC3, which display a canonical bivalent chromatin state that is 'poised' for transcription (Fig. EV2B) (Antunes et al, 2024; Farhat et al, 2020). These promoters are structurally poised for activation prior to stage conversion. The degradation of MORC keeps the chromatin open, making DNA available to secondary factors, which guide the differentiation towards either the merozoite or bradyzoite stage (Fig. EV2B) (Pachano et al, 2025).

## *BSM*-deficient parasites exhibit no growth defect in vitro and maintain chronic infection capability in mice

BSM does not have a significant effect on tachyzoite proliferation and survival in culture, as *bsm*-deficient parasites were able to grow and form plaques in vitro similarly to the parental strain (Fig. EV2C). To further assess the impact of *BSM* knockout in vivo, $5.10^4$ tachyzoites from the 76 KΔ*bsm* parasite line were inoculated intraperitoneally into a group of 8 NMRI mice, with a control group of 8 NMRI mice infected with the 76 K wild-type strain. Both groups showed similar clinical symptoms, including weight loss and ruffled fur. One mouse infected with the 76KΔ*bsm* parasite died,

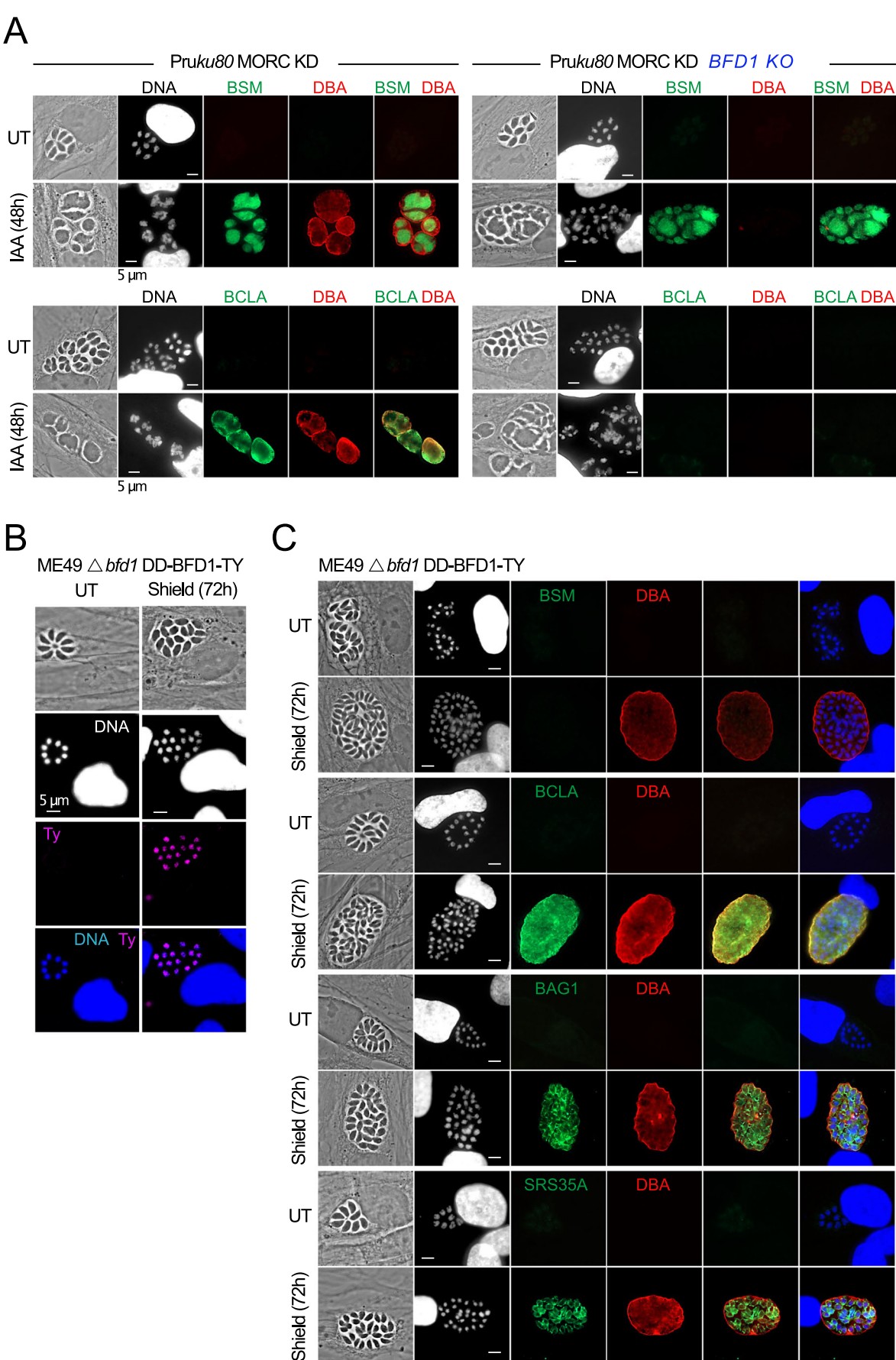

◀ **Figure 3. MORC/HDAC3 operates upstream of BFD1, which regulates BCLA expression but not BSM.**

(A) IFA of BSM (in green, top panel) and BCLA (in green, bottom panel) expression in wild-type and *BFD1* KO strains under a MORC KD background, either untreated or treated with IAA for 48 h. The cells were co-stained with DBA lectin (red) and DNA-specific Hoechst dye (white). Scale bar = 5 μm. (B) IFA of BFD1 (red) in a strain expressing a regulatable BFD1 protein (ΔBFD1/DD-BFD1-Ty) with and without Shield-1 treatment. Cells were co-stained with DNA-specific Hoechst dye (white or blue). Scale bar = 5 μm. (C) IFA of BSM, BCLA, BAG1, and SRS35A expression (green) in ΔBFD1/DD-BFD1-Ty strains, either untreated or stimulated with Shield-1 for 72 h. Cells were co-stained with DBA lectin (red) and Hoechst dye (white or blue) for DNA visualization. Scale bar = 5 μm. Source data are available online for this figure.

but overall survival rates over two months were similar, showing no significant difference in strain virulence (Mantel–Cox test and Gehan–Breslow-Wilcoxon test $P = 0.3496$) (Fig. 5A). All mice showed a strong adaptive immune response, as shown by western blots (Fig. EV2D), except one that was likely poorly infected during parasite inoculation and was therefore excluded from the study. Cystogenesis was unaffected in mice infected with *bsm*-deficient parasites, with cysts displaying the same round morphology as the parental strain (Fig. 5B). However, mice infected with 76KΔ*bsm* showed a significantly higher parasite load in their brains (Fig. 5C), along with increased expression miR-146a and miR-155 (Fig. 5D), two host microRNAs associated with *Toxoplasma* persistence in the brain (Cannella et al, 2014). This increased parasite burden, which may have been overlooked by Yang et al due to the shorter time period in their study (30 days versus two months in our study; Yang et al, 2017), raises the possibility that BSM may play a role in modulating parasite persistence. However, additional analyses, including direct quantification of cysts, would be required to clarify the nature of this increased parasite burden and whether it reflects a difference in cyst load or potential parasite reactivation.

## Bradyzoite serology with BSM and BCLA proxies accurately identifies mice harboring cysts

BSM and BCLA, both identified as potential biomarkers for chronic toxoplasmosis, are located in distinct cellular sites: BSM within the bradyzoite's cytoplasm, and BCLA on the cyst surface. This difference in subcellular localization could impact how they're presented as antigens and influence antibody production. Therefore, we aimed to directly compare the immunogenicity of BSM and BCLA side-by-side. We produced the recombinant BSM protein tagged with a C-terminal histidine tag using the baculovirus expression system. Unlike expression in prokaryotic cells, eukaryotic systems produce proteins surface post-translational modifications that more closely mimic those of the native protein (Chambers et al, 2018). Additionally, they offer the benefit of co-purifying fewer immunogenic contaminants typically associated with *E. coli*-based expression strategies. After affinity purification on Ni-NTA resin (Fig. EV3A) and subsequent size exclusion chromatography, we obtained highly purified recombinant BSM protein in substantial quantities (Fig. EV3B). The protein behaves as a monomer in solution, allowing the evaluation of its immunogenicity. Concurrently, to improve our original ELISA assay against BCLA (Dard et al, 2021), we expressed in insect cells a new internally truncated version of BCLA that includes all the immunogenic regions of the protein (Fig. EV3C). This revised version of the recombinant protein is natively soluble, stable, and straightforward to purify, unlike the bacterially produced C-terminus, which required partial denaturation for solubilization (Dard et al, 2021). It includes both the N-terminal start, a

minimized immunogenic repeat regions with two full repeats, and the predicted structured C-terminus (Fig. EV3C). The protein has a predicted molecular weight of 75.6 kDa and migrates accordingly on an SDS-PAGE gel (Fig. EV3C,D). However, the size exclusion chromatography profile indicates a tendency for the complex to form higher-order oligomers or soluble aggregates (Fig. EV3D). Previously, ELISAs were performed with a partial fragment of BCLA produced in bacteria alongside synthetic peptides (Dard et al, 2021); this new strategy has standardized the antigen production process in native purification conditions (Fig. EV3C). Subsequently, an ELISA assay was developed and used to measure anti-BSM and anti-BCLA antibody titers in sera from 83 NMRI, CD1, and Balb/c mice, either uninfected or infected with different parasite strains (Fig. 5E,F).

BSM serology specifically identified mice infected with cystogenic strains (76 K and ME49) that exhibited positive cerebral parasite loads, except for mice infected with the 76 KΔ*bsm* strain, underscoring the specificity of the assay (Fig. 5F). In contrast, mice infected with strains characterized with low cystogenic potential (culture-attenuated PruΔ*Ku80* and CTG) and had very low or undetectable cerebral parasite loads exhibited negligible or no anti-BSM titers (Fig. 5E). Similarly, BCLA serology effectively identified cyst-bearing mice, including those infected with the 76 KΔ*bsm* strain but not those infected with non-cystogenic strains (Fig. 5F). However, the performance of BCLA antigen in discriminating cyst carriage was slightly reduced compared to the BSM antigen (area under the ROC curves: 0.9256 (IC$_{95}$: 0.8651–0.9861) and 0.9923 (IC$_{95}$: 0.9765–1), respectively, $P = 0.0365$) (Hanley and McNeil, 1983) (Fig. 5G). The ELISA cutoff values were set at 15.85 and 3.286 UI for BSM and BCLA, respectively, optimized for specificity. The sensitivity and specificity of BSM ELISA were 97.96% (IC$_{95}$: 89.31–99.90) and 100.0% (IC$_{95}$: 86.20–100.0), respectively, whereas the BCLA ELISA demonstrated a sensitivity of 76.47% (IC$_{95}$: 63.24–86.00) and a specificity of 100.0% (IC$_{95}$: 79.61–100.0) (Fig. 5G). The agreement between the two serological tests was quantified by a *kappa* coefficient of 0.736, corresponding to substantial agreement (Landis and Koch, 1977). In conclusion, although the BSM ELISA proved more effective than BCLA in identifying cystogenic infections, both biomarkers offer reliable tools that can be combined to detect cyst carriage in mouse models.

## Bradyzoite-specific serology in human patients with different forms of toxoplasmosis

The next challenge was to determine whether BSM can be validated as a serological biomarker for detecting cysts during chronic infection in humans. To address this, we performed side-by-side ELISA assays to compare the reactivity of BCLA and BSM with human sera from two French toxoplasmosis biobanks (Fig. 6A,B). Initially, we compared bradyzoite serology results between two

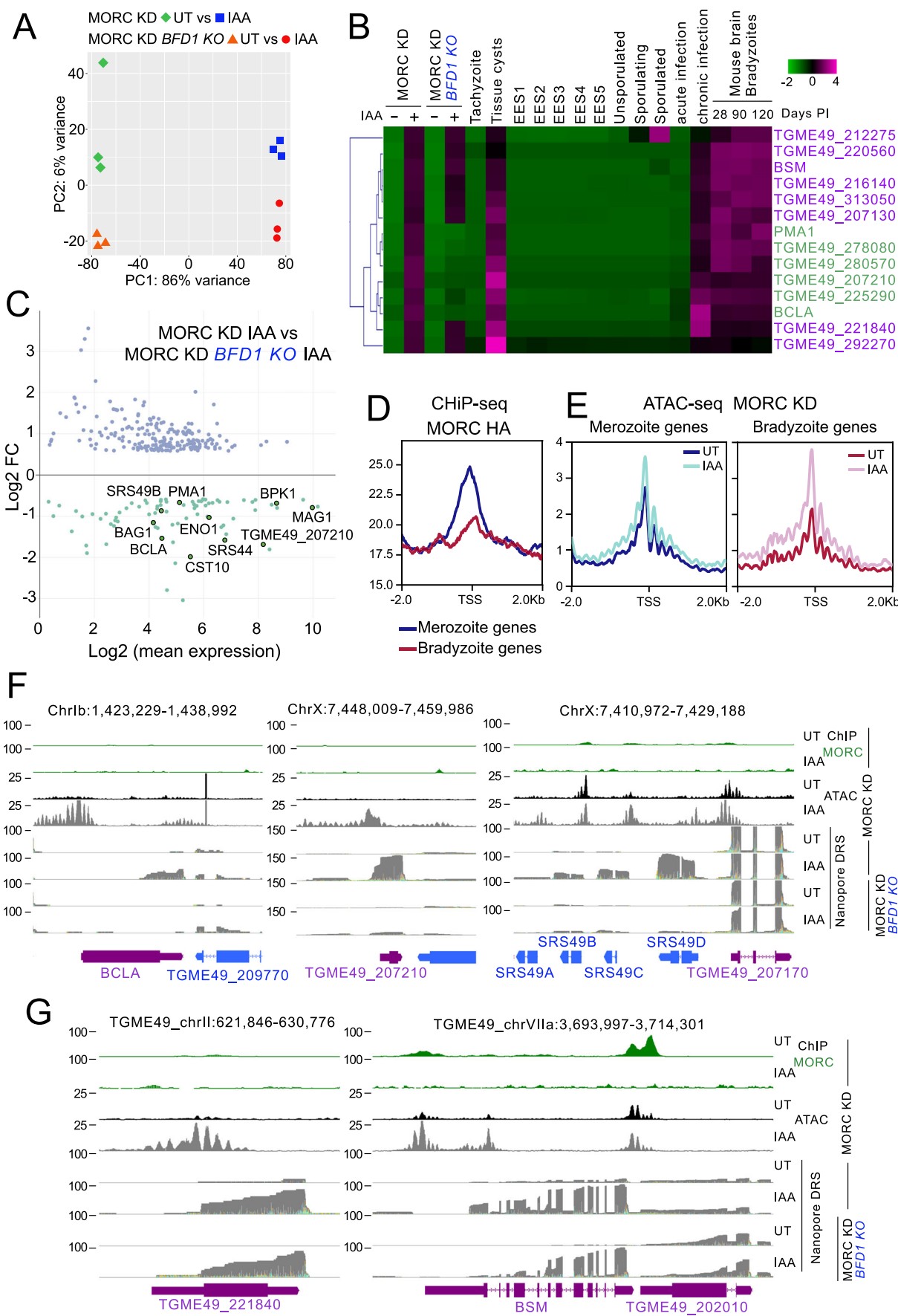

◀  **Figure 4.  MORC exerts epistatic control over BFD1, which regulates a subset of the bradyzoite transcriptome.**

(A) Principal Component Analysis (PCA) of mRNA sequencing data from biological triplicates of MORC KD or MORC KD/*BFD1* KO parasites. Samples were collected from untreated conditions or after 24 h of IAA treatment. (B) Heatmap showing hierarchical clustering analysis of selected bradyzoite-specific mRNA transcripts, which were significantly upregulated (log2[FC] > 2; *P* value < 0.05) following the depletion MORC in both MORC KD or MORC KD/*BFD1* KO genetic background. The abundance of these transcripts is presented across different in vivo stages—merozoites, EES1–EES5 stages, tachyzoites, sporozoites and cysts—as documented in previous studies (Antunes et al, 2024; Farhat et al, 2020). Pertinent examples of BFD1-dependent and independent genes are highlighted in green and violet, respectively. (C) MA plots display Log2(FC) against Log2(mean expression) for genes comparing MORC KD to MORC KD/*BFD1* KO after IAA treatment, using DESeq2. BFD1-dependent genes are shown in green. (D) ChIP-seq. (E) ATAC-seq. (F, G) Integrated Genome Browser (IGB) screenshots of representative bradyzoite genes displaying ChIP-seq signal for MORC (HA antibody, green) in MORC KD strains under untreated and IAA-treated conditions. ATAC-seq profiles for both conditions, showing Tn5 transposase accessibility with read density on the *y* axis, are included. Nanopore DRS data for MORC KD and MORC KD/*BFD1* KO strains, untreated and IAA-treated, are also shown. Examples include BFD1-dependent (F) and BFD1-independent genes (G).

groups: individuals with negative conventional serology (*n* = 134) and those previously identified as positive (*n* = 222). While antibody titers drop below the detection limit of conventional serology over time in some subjects (Burrells et al, 2016; Rougier et al, 2017), it is unclear if this correlates with cyst disappearance. Positive tachyzoite serology does not always indicate cyst presence either. However, it is reasonable to hypothesize that *T. gondii* cysts are frequently present in individuals with positive conventional serology, though their quantity may vary between individuals. In contrast, they are likely to be rare or absent in seronegative individuals. This forms the basis for interpreting *Toxoplasma* serology in clinical laboratories worldwide. As expected, the "past immunity" group showed significantly higher anti-BCLA and anti-BSM antibody titers. For BCLA ELISA, medians and percentiles were 9.87 (5.15–16.9) in the seronegative group versus 20.02 (12.6–42.2) in the "past immunity" group. For BSM ELISA, the index was 0.945 (0.730–1.41) versus 1.30 (0.875–2.05) (*P* < 0.0001 for both tests) (Fig. 6A,B). However, results in the "past immunity" group were widely scattered, with several patients showing low titers (Fig. 6A, B). ROC curves were generated using these serological results to determine optimal thresholds (Fig. 6C). The cutoff values were set at 34.52 UI for BLCA and 1.697 (index ratio) for BSM. These thresholds allowed identification of patients with past immunity with a sensitivity and specificity of 31.08% (IC$_{95}$: 25.36–37.45) and 97.76% (IC$_{95}$: 93.62–99.39) for BCLA and 32.43% (IC$_{95}$: 26.62–38.84) and 83.58% (IC$_{95}$: 76.39–88.90) for BSM, respectively (Fig. 6C).

Within the "past immunity" group, we focused on clinical contexts commonly associated with cyst reactivation: ocular toxoplasmosis (*n* = 23), cerebral toxoplasmosis (*n* = 5), and asymptomatic reactivations identified through serological screening of immunocompromised patients (*n* = 6). Although patient numbers were limited in these subgroups, bradyzoite serology titers were notably higher than those in the seronegative group (Fig. 6A,B). Agreement between the two tests, measured by Cohen's kappa coefficient, was 0.312, indicating "fair agreement" (Landis and Koch, 1977). Notably, while only 17.16% of the seronegative group had at least one positive bradyzoite serology test, this proportion rose to 46.84% in the "past immunity" group (Dataset EV2), suggesting individual variability in response to each marker rather than consistent detection of both markers. In addition, nine patients with clinical conditions typically linked to cyst reactivation tested negative for both antigens (Datasets EV3 and EV4). In summary, BSM is a promising serological marker for chronic *T. gondii* infection in humans, with high specificity for detecting past immunity. Although BSM shows "fair agreement" with BCLA and

some variability, it may serve as a potential marker for cyst reactivation in conditions such as ocular and cerebral toxoplasmosis.

## Assessing the effectiveness of cyst and bradyzoite serology in differentiating acute and past infections and identifying congenital infections

To investigate the utility of bradyzoite serology in differentiating acute and past infections, we measured the titers of anti-BSM and anti-BCLA antibodies in a study of 115 iterative serum samples from 39 pregnant women who seroconverted during pregnancy and one man with acute infection. Of these, infections from 74 sera were precisely dated to within 4 months post-infection and classified as the "acute infection" group. We compared these titers with samples from a "past immunity" group, where IgG titers on the *Architect ®Toxo IgG* assay exceeded 20 UI/ml, indicating a sustained immune response. It was assumed that some patients with lower reactivity to tachyzoites may have been infected too long ago, potentially having eliminated their cysts and bradyzoites. Statistically significant differences in anti-BSM and anti-BCLA antibody titers were observed between the "acute infection" and "past infection" groups (*P* = 0.0303 and 0.0009, respectively) (Fig. 6D; Dataset EV4). These results support the potential role of bradyzoite-based serology in maternal screening by providing a tool to distinguish between acute and past infections, addressing one of the challenges in congenital toxoplasmosis prevention (van der Giessen et al, 2021). Such a differentiation is critical, especially when serologic profiles are atypical or IgM persists, complicating the dating of infections and subsequent assessment of fetal risk (Fricker-Hidalgo et al, 2013; Gras et al, 2004).

A major challenge in diagnosing congenital toxoplasmosis is that maternal IgG can cross the placenta, and the absence of IgM in newborns does not rule out infection. To address this, we evaluated BSM and BCLA serology at birth in children born to mothers who seroconverted during pregnancy. Among the children, 15 were diagnosed with congenital toxoplasmosis, and 11 were not, based on thorough post-natal monitoring in their first year. While BSM serology showed no significant difference between the groups (*P* = 0.5063), BCLA serology was significantly elevated in the congenital toxoplasmosis group (*P* = 0.0077) (Fig. 6E; Dataset EV4). In summary, BCLA serology at birth shows promise as a diagnostic tool for congenital toxoplasmosis, distinguishing infected from non-infected infants born to mothers who seroconverted during pregnancy. This marker could enhance early diagnosis and aid in the prompt intervention for affected newborns.

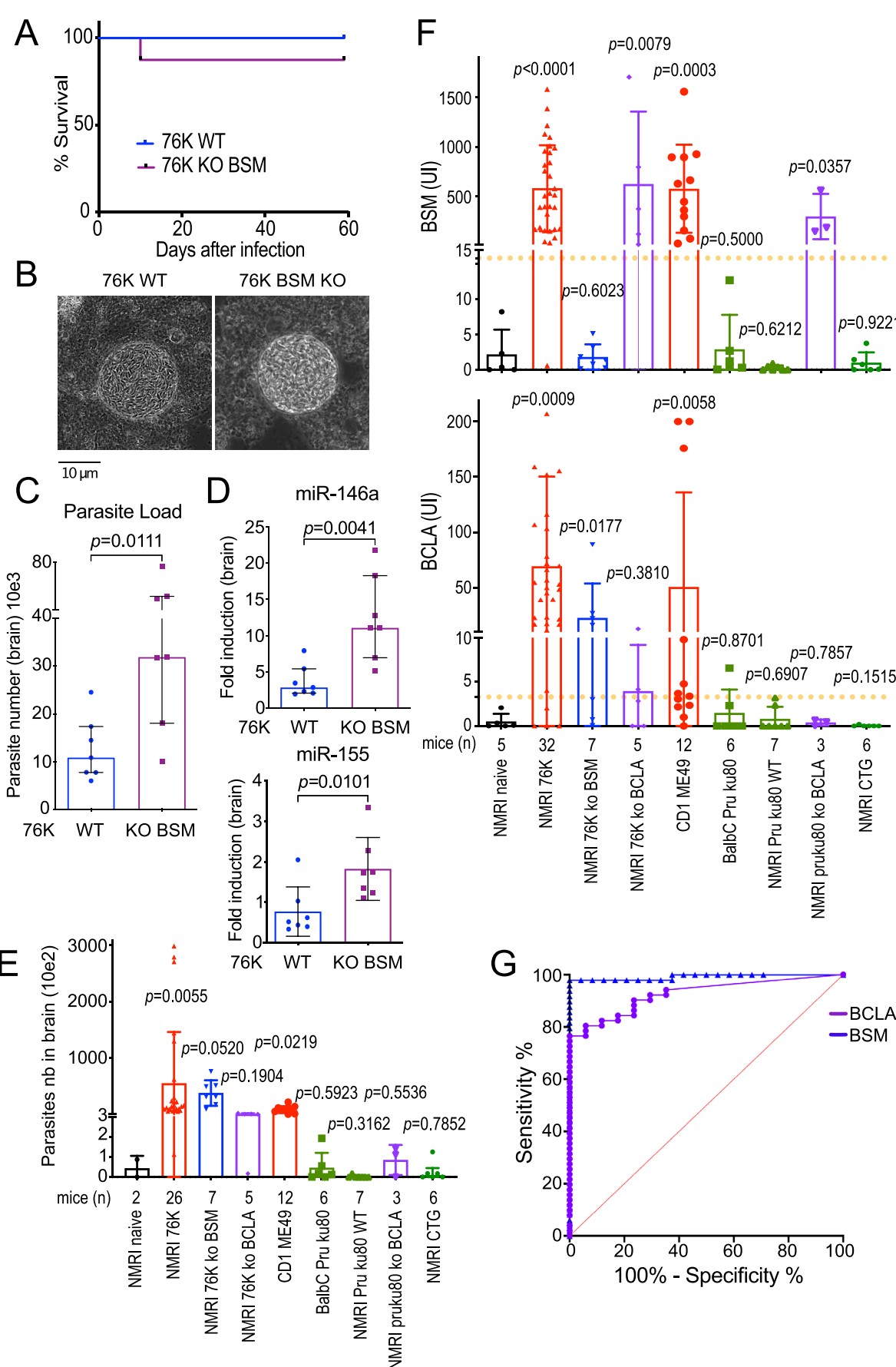

◄

**Figure 5. Bradyzoite serology with BSM and BCLA accurately detects cyst-carrying mice.**

(A) NMRI mice ($n = 15$) were injected intraperitoneally with $5.10^4$ tachyzoites of the 76K-GFP-luc WT or $\Delta bsm$ strains, and their survival was monitored. Statistical analysis using log-rank Mantel–Cox and Gehan–Breslow–Wilcoxon tests showed no significant difference ($P = 0.3496$). (B) Microscopy analysis of cyst morphology in homogenized brain suspensions from mice infected with 76K-GFP-luc and $\Delta bsm$ strains. Over 20 cysts were examined per strain, and representative images are presented. (C, D) Parasitic loads in parasite per brain (qPCR count) and expression levels of miR-155 and miR-146a were assessed in mice infected by 76 K WT ($n = 7$) versus 76 K $\Delta bsm$ ($n = 7$). Statistical significance was calculated using a nonparametric Mann–Whitney test. (E) Parasitic loads (parasites per brain, quantified by qPCR) were measured in mice ($n = 74$) of various backgrounds (NMRI, Balb/C, CD1) infected with cystogenic strains (76 K, ME49; in red), low cystogenic strains (CTG, Pruku80; in green), 76 K $\Delta bsm$ (in blue), and 76 K or Pruku80 $\Delta bcla$ (in purple). (F) BSM and BCLA ELISA serology in mice ($n = 83$) is shown using the same color coding as in panel (E). Statistical significance was calculated using a nonparametric Mann–Whitney test. (G) ROC (Receiver Operating Characteristic) curves for BSM ELISA serology (blue) and BCLA ELISA serology (purple). ROC curves are graphical representations that evaluate the diagnostic performance of a test by plotting the true positive rate (sensitivity) against the false positive rate (1 − specificity) at various thresholds. (C–F) The error bars indicate the standard error of the mean (SEM). Source data are available online for this figure.

## Discussion

With a global seroprevalence estimated at 30% (Pappas et al, 2009) and up to 50% in some regions (Wilking et al, 2016), *T. gondii* is believed to reside in the brains of more than 2 billion individuals worldwide. Improving diagnostics is critical to addressing this health burden (Ben-Harari and Connolly, 2019; Cantey et al, 2021), especially since the current standard therapy (pyrimethamine and sulfadiazine) has notable side effects and cannot eliminate chronic infections due to the inaccessibility of tissue cysts. While meat-borne transmission is a major route of infection, current surveillance and control strategies are hindered by the lack of robust diagnostic tools (Bouwknegt et al, 2018; Kijlstra and Jongert, 2009). Conventional serological assays, relying on antibodies against acute-phase antigens, frequently fall short in reliably detecting chronic infections (Kijlstra and Jongert, 2009). Historically, serological tests have focused on tachyzoite antigens due to their ease of culture and high immunogenicity (Zhang et al, 1995). However, recent studies overturn the long-standing belief that cyst antigens are weakly immunogenic (Roiko et al, 2018). The discovery of BCLA, a cyst-specific protein, revealed strong immunogenic properties, opening new possibilities for serological markers (Dard et al, 2021). Our present study expands on this by identifying BSM, another cyst antigen with unexpected immunogenicity, despite its internal location within bradyzoites, which typically limits immune exposure. Unlike cyst surface-exposed BCLA, BSM is theoretically only accessible to the immune system when bradyzoites break down, raising intriguing questions about antigen presentation—whether it occurs within neurons or at peripheral sites.

Our findings also highlight the intricate regulatory dynamics between MORC/HDAC3 and BFD1 in bradyzoite development and point to an emerging complexity in antigenic potential among bradyzoite-specific proteins. While MORC/HDAC3 restricts gene expression through chromatin accessibility, BFD1 selectively regulates bradyzoite-specific genes, excluding BSM, which appears independently controlled. This regulatory distinction supports a nuanced view of bradyzoite immunogenicity, where BSM and BCLA play different roles in immune recognition. This underscores the need for a broad, unbiased exploration of bradyzoite proteins to uncover potential serologic markers.

We were able to show that BSM serology is consistent with BCLA serology and reliably correlates with parasitic DNA in the brains of chronically infected mice. Moreover, this correlation extends specifically to cyst burden in mice, with miR-155 and miR-146a serving as effective proxies for the presence of cysts (Cannella et al, 2014). In this regard, bradyzoite serology can be used as a non-lethal method to detect cysts in *T. gondii* studies in mice and represents a significant ethical advance as it reduces the use of animals in longitudinal research. Testing this serology in farm animals could also determine its accuracy in assessing meat infectivity and potentially pave the way for "*Toxoplasma*-free" meat production.

Bradyzoite ELISA serologies in humans reveal a more complex scenario. Individuals with long-lasting immunity, as indicated by conventional tachyzoite serology, generally show higher BSM and BCLA titers compared to *T. gondii*-negative individuals. However, significant variability in both groups raises questions about the reliability of positive bradyzoite serology in those considered seronegative by conventional methods and the relevance of negative bradyzoite serology in individuals with a known infection history. While it remains unclear whether cysts persist for life in all previously infected individuals, bradyzoite serology in humans lacks full sensitivity. In a small group of patients with cyst-associated clinical disease, bradyzoite serology did not detect all cases, even when conventional serology was positive. However, it shows promise as a complementary tool for managing complex cases in specialized reference centers.

In pregnancy, bradyzoite serology could play a critical role in distinguishing recent from past infections (Chemla et al, 2002; Villena et al, 1998), with higher BSM and BCLA titers linked to chronic infections. Yet, specificity remains modest, and the kinetics of cystogenesis in humans are poorly understood, with bradyzoite formation possibly appearing earlier than assumed. Notably, BCLA serology—unlike BSM—was elevated at birth in congenital toxoplasmosis cases, reinforcing its diagnostic value, especially for early detection. With the discontinuation of ISAGA, formerly considered a gold-standard technique for diagnosing congenital toxoplasmosis, this presents an opportunity to replenish the toolbox for diagnosing this pathology (Montoya, 2024).

Interpreting human results remains challenging due to the absence of a definitive reference for cyst presence, making it difficult to distinguish between serology imprecision and unknown cystogenesis aspects. Nonetheless, bradyzoite serology, by better targeting latent infections, could offer crucial insights into controversies surrounding *T. gondii*'s effects on human behavior—long documented in rodents (Boillat et al, 2020; Dupont et al, 2021) but debated in human studies.

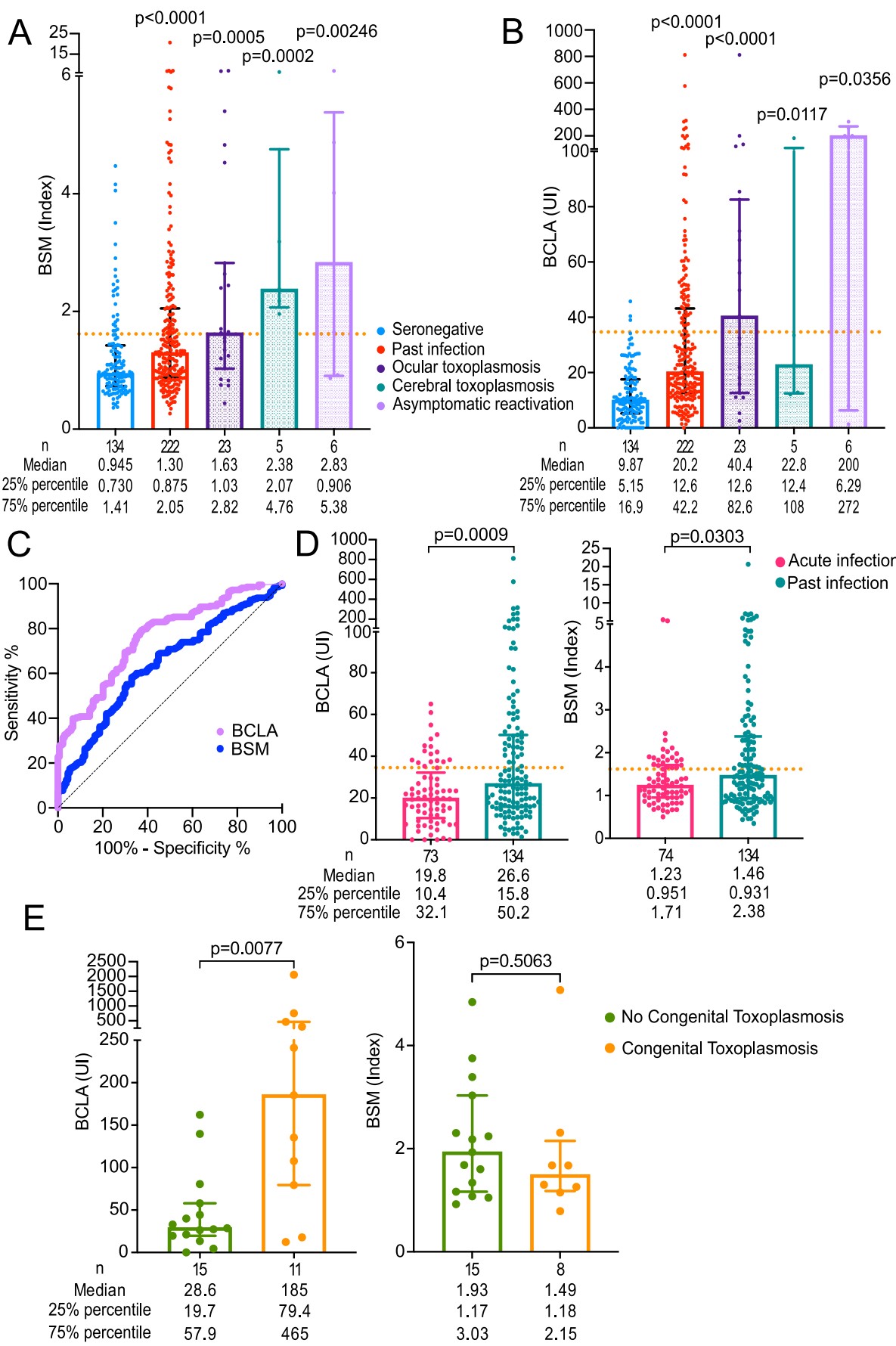

**Figure 6. Bradyzoite-specific serology in human sera.**

(A) BSM ELISA serology and (B) BCLA ELISA serology results in seronegative individuals ($n = 134$, blue bars) and previously immunized patients with *T. gondii* infection ($n = 222$, red bars), grouped by clinical context (grid-patterned bars). Statistical significance was evaluated using a nonparametric Mann–Whitney test. (C) Receiver operating characteristic (ROC) curves for BSM ELISA (blue) and BCLA ELISA (purple), illustrating diagnostic performance. (D) BSM and BCLA ELISA serology titers during past infections (pink bars) and acute infections (green bars). Statistical comparisons were performed using a nonparametric Mann–Whitney test. (E) BSM and BCLA ELISA serology titers in newborns from mothers who seroconverted during pregnancy. Results are shown for children diagnosed with congenital toxoplasmosis (CT, $n = 15$, green bars) and those where CT was excluded ($n = 11$, orange bars). Statistical significance was calculated using a nonparametric Mann–Whitney test. Source data are available online for this figure.

# Methods

### Reagents and tools table

| Reagent/resource | Reference or source | Identifier or catalog number |
|---|---|---|
| **Biological samples and experimental models** | | |
| NMRI (*M. musculus*) | Janvier Laboratories | RjHan:NMRI |
| CD1 (*M. musculus*) | Charles River | CD1® IGS Mouse |
| Balb/cJRJ (*M. musculus*) | Janvier Laboratories | BALB/cByJRj |
| RHku80 MORC KD | Farhat et al, 2020 | |
| Pru*ku80* MORC KD | Farhat et al, 2020 | |
| RH*ku80* MORC KD BSM-Flag | This study | |
| RH*ku80* MORC KD TGME49_216140-Flag | This study | |
| 76K-luc-GFP | Dard et al, 2021 | |
| 76K-luc-GFP *bsm* KO | This study | |
| ME49 | Antunes et al, 2024 | |
| CTG | Cannella et al, 2014 | |
| Pru*ku80* | Antunes et al, 2024 | |
| Pruku80 *bcla* KO | Dard et al, 2021 | |
| 76K-luc-GFP *bcla* KO | Dard et al, 2021 | |
| Pruku80 BFD1-HF | | |
| RHku80 MORC KD BFD1-Flag | This study | |
| Pruku80 MORC KD/*BFD1* KO | This study | |
| ME49 *bfd1* KO DD-BFD1-TY | Waldman et al, 2020 | |
| SF21 cells | Cliniciences | 94-003F |
| Hi-5 cells | Cliniciences | 94-002F |
| Human primary fibroblasts | ATCC | Cat# SCRC-1041 |
| EMBacY baculovirus | gift from I. Berger | |
| Patient serums | Grenoble Alpes University Hospital and Hospices Civils de Lyon | |
| **Recombinant DNA** | | |
| Plasmid pDHFR-TSc3 | Antunes et al, 2024 | |
| Plasmid pLIC-HF-dhfr | Antunes et al, 2024 | |
| Plasmid pTOXO_Cas9-CRISPR | Antunes et al, 2024 | |
| Plasmid pTOXO_Cas9-CRISPR::sgBSM | This study | |
| Plasmid pLIC-BSM-HF | This study | |
| Plasmid pLIC-TGME49_216140-HF | This study | |
| TgBSM construct | This study, GenScript | |
| TgBCLA construct | This study, GenScript | |

| Reagent/resource | Reference or source | Identifier or catalog number |
| --- | --- | --- |
| pFastBac1 vector | Invitrogen | |
| **Antibodies** | | |
| Rabbit anti-TgHDAC3 | Eurogentec custom antibody | RRID: AB_2713903 |
| Mouse anti-TgBAG1 | Farhat et al, 2020 | |
| Mouse anti-TgGAP45 | gift from Pr. D. Soldati | |
| Mouse anti-HA | Roche | RRID: AB_2314622 |
| Rabbit anti-HA | Cell Signaling Technology | RRID: AB_1549585 |
| Rabbit polyclonal anti-QRS | Antunes et al, 2024 | |
| Rabbit polyclonal anti-BCLA | Dard et al, 2021 | |
| Mouse anti-FLAG | Cell Signaling Technology | Cat#8146 |
| Rabbit anti-FLAG | Cell Signaling Technology | RRID: AB_2798687 |
| Rabbit polyclonal anti-BSM | Eurogentec custom antibody | |
| Alexa Fluor 488 Recombinant Rabbit Monoclonal Antibody | Invitrogen | A11008 |
| Alexa Fluor 488 Recombinant Mouse Monoclonal Antibody | Invitrogen | A11001 |
| Alexa Fluor 594 Recombinant Rabbit Monoclonal Antibody | Invitrogen | A11012 |
| Alexa Fluor 594 Recombinant Mouse Monoclonal Antibody | Invitrogen | A11005 |
| Anti-Mouse IgG, AP Conjugate | Promega | S3721 |
| Anti-Rabbit IgG, AP Conjugate | Promega | S3731 |
| Anti-His-tag chimeric human monoclonal antibody | Sigma-Aldrich | SAB5600096 |
| Anti-Mouse IgG (Fc specific)–Peroxidase antibody | Sigma-Aldrich | Cat# A0168 |
| Anti-Human IgG (Fc specific)—Peroxidase antibody | Sigma-Aldrich | Cat# A0170 |
| **Oligonucleotides and other sequence-based reagents** | | |
| Oligonucleotide LIC-216140-Fwd | Sigma-Aldrich | TACTTCCAATCCAATTTAGCtccgaagaccatccatgaatattcatgg |
| Oligonucleotide LIC-216140-Rev | Sigma-Aldrich | TCCTCCACTTCCAATTTTAGCgccgtttatctcgaccacggatggcgg |
| Oligonucleotide LIC-BSM-Fwd | Sigma-Aldrich | TACTTCCAATCCAATTTAGCgatgaattaccccacagtgtgtggac |
| Oligonucleotide LIC-BSM-Rev | Sigma-Aldrich | TCCTCCACTTCCAATTTTAGCagctgtgtgagaatgctgccgctcgg |
| Oligonucleotide BSM-CRISP1-Fwd | Sigma-Aldrich | AAGTTGATGATACTCACTGCAGCTCG |
| Oligonucleotide BSM-CRISP1-Rev | Sigma-Aldrich | AAAACGAGCTGCAGTGAGTATCATCA |
| Oligonucleotide BSM-REV1-screen | Sigma-Aldrich | GAGCTGCAGTGAGTATCATC |
| Oligonucleotide BSM-CRISP2-Fwd | Sigma-Aldrich | AAGTTGGAGGAAACTTTCAGCAAAGG |
| Oligonucleotide BSM-CRISP2-Rev | Sigma-Aldrich | AAAACCTTTGCTGAAAGTTTCCTCCA |
| Oligonucleotide BSM-REV2-screen | Sigma-Aldrich | CTTTGCTGAAAGTTTCCTCC |
| **Chemicals, enzymes, and other reagents** | | |
| Dulbecco's modified Eagle's medium | Invitrogen | Cat# 12491015 |
| Fetal Bovine Serum | Gibco | 10270106 |
| HEPES | Gibco | Cat# 15630056 |
| L-glutamine | Gibco | 25030123 |
| Pénicilline-streptomycine | Gibco | 15070063 |
| Hank's Balanced Salt Solution | Gibco | Cat# 14025-050 |
| 3-indoleacetic acid | Sigma-Aldrich | Cat# 45533 |

| Reagent/resource | Reference or source | Identifier or catalog number |
| --- | --- | --- |
| FR235222 | Antunes et al, 2024 | |
| DMSO | Sigma-Aldrich | D2438 |
| Bovine serum albumin | Sigma-Aldrich | A7030 |
| Hoechst 33342 solution | Invitrogen | Cat# H1399 |
| Complete EDTA free | Roche | 05056489001 |
| NBT/BCIP 1-Step | Thermo Fisher Scientific | Cat# 34042 |
| Coomassie blue R-250 | Sigma-Aldrich (MERK) | 1.12553.0025 |
| TRIzol | Thermo Fisher Scientific | Cat# 15596026 |
| Pyrimethamine | Sigma-Aldrich (MERK) | 46706 |
| ESF921 media | Expression System | Cat # 96-001-01 |
| Benzonase | MERK Millipore | Cat # 70746 |
| Ni-NTA resin | Qiagen | Cat# 30210 |
| SuperBlock blocking buffer | Thermo Fisher Scientific | Cat# 37515 |
| Recombinant protein A/G peroxidase conjugated | Thermo Fisher Scientific | Cat# 32490 |
| TMB Substrate | Thermo Fisher Scientific | Cat# 34029 |
| **Software** | | |
| ZEN 2 lite 9 | Carl Zeiss | |
| GraphPad Prism 8 | https://www.graphpad.com/features | |
| Xcalibur 2.8 | Thermo Fisher Scientific | |
| Mascot v2.8.3 | Matrix Science | |
| Proline v2.2.0 | ProFI Proteomics | |
| Astra | Wyatt Technologies | |
| FastQC | www.bioinformatics.babraham.ac.uk/projects/fastqc/ | |
| MultiQC | Seqera | |
| iDEP.96 | http://bioinformatics.sdstate.edu/idep96/ | |
| Minimap2 | https://github.com/lh3/minimap2 | |
| GuppY v4.09 | | |
| Integrated Genome Browser (IGB) v9.1.8 | https://www.bioviz.org/ | |
| Deeptools v3.5.1 | https://deeptools.readthedocs.io/en/3.5.2/index.html | |
| BOWTIE2 software (V2.1.0) | https://github.com/ | |
| MACS v2.2 | https://github.com/ | |
| nf-core Chip-Seq v2.0.0 | https://github.com/ | |
| Samtools v1.4 | https://www.htslib.org/ | |
| Bamtools v2.5.1 | https://github.com/ | |
| **Other** | | |
| QiAamp DNA mini kit | Qiagen | Cat# 51304 |
| TaqMan™ MicroRNA Reverse Transcription Kit | Applied Biosystems | Cat# 4366596 |
| ABI 7500 real-time PCR system | Applied Biosystems | |
| Ultimate 3000 RSLCnano | Thermo Fisher Scientific | |
| Q-Exactive Plus | Thermo Fisher Scientific | |
| BTX ECM 630 | Harvard Apparatus | |
| ÄKTA pure™ chromatography system | Cytiva | |

| Reagent/resource | Reference or source | Identifier or catalog number |
|---|---|---|
| Amicon ultra concentrator | Sigma-Aldrich | |
| Mono-Q column | Cytiva | |
| Superdex S200 column | Cytiva | |
| L-2400 UV detector | Hitachi | |
| Optilab T-rEX refractometer | Wyatt technologies | |
| DAWN HELEOS-II multi-angle light scattering detector | Wyatt technologies | |
| Medisorp plates | Nunc | Cat# 467320 |
| Gemini ELISA automation platform | Stratec | |
| RNeasy Plus Mini Kit | Qiagen | Cat# 74134 |
| NanoDrop 2000 | Thermo Fisher Scientific | |
| Agilent 5400 Fragment Analyzer System | Agilent Technologies | |
| Illumina NovaSeq platform | Illumina | |
| Qubit HS dsDNA | Thermo Fisher Scientific | Cat# Q32851 |
| minION sequencer | Oxford Nanopore Technologies | |
| FLO-MIN106D flow cells | Oxford Nanopore Technologies | |
| Architect platform | Abbott | |
| VIDAS 3 | BioMérieux | |
| Architect Toxo IgG | Abbott | |
| Architect Toxo IgM | Abbott | |
| VIDAS® TOXO IgG II | BioMérieux | Cat# 30210-01 |
| VIDAS® TOXO IgM | BioMérieux | Cat# 30202-01 |
| TOXOPLASMA WB IgG | LDBIO Diagnostics | Cat #TOP-WB12GM |
| NuPAGE 4–12% | Invitrogen | Cat# NP0323BOX |
| Fluorescence microscope ZEISS ApoTome.2 | Carl Zeiss | |

## Parasites and human cell culture

Human primary fibroblasts (HFFs, ATCC® CCL-171™) were cultured in Dulbecco's modified Eagle's medium (DMEM) (Invitrogen) supplemented with 10% heat-inactivated fetal bovine serum (FBS) (Invitrogen), 10 mM (4-(2-hydroxyethyl)-1-piperazine ethane sulphonic acid) (HEPES) buffer pH 7.2, 2 mM L-glutamine, and 50 μg/mL of penicillin and streptomycin (Invitrogen). Cells were incubated at 37 °C in 5% $CO_2$. The *Toxoplasma* strains were maintained in vitro by serial passage on monolayers of HFFs.

## Reagents

The following primary antibodies were used in the immunofluorescence (1:1000 dilution), and immunoblotting assays (1:1000 dilution): rabbit anti-TgHDAC3 (RRID: AB_2713903), mouse anti-TgBAG1, mouse anti-HA tag (Roche, RRID: AB_2314622), anti-QRS (Glutaminyl-tRNA synthetase), rabbit anti-BCLA (Dard et al, 2021). Immunofluorescence secondary antibodies were coupled with Alexa Fluor 488 or Alexa Fluor 594 (Thermo Fisher Scientific) and used at 1:1000 dilution. Secondary antibodies used in western blotting were conjugated to alkaline phosphatase (Promega) and used at 1:5000 dilution. We also commissioned Eurogentec to produce polyclonal serum from rabbits immunized against the full-length BSM recombinant protein and used it for immunofluorescence and immunoblotting assays.

## Mouse infection and experimental survey

Six-week-old NMRI, CD1, or Balb/C mice were obtained from Janvier Laboratories (Le Genest-Saint-Isle, France). Mouse care and experimental procedures were performed under pathogen-free conditions in accordance with established institutional guidance and approved protocols from the Institutional Animal Care and Use Committee of the University Grenoble Alpes (APAFIS#4536-2016031 017075121 v5). Female mice were used for all studies. For intraperitoneal (i.p.) infection; tachyzoites were grown in vitro and extracted from host cells by passage through a 27-gauge needle, washed three times in phosphate-buffered saline (PBS), and quantified with a hemocytometer. Parasites were diluted in Hank's Balanced Salt Solution (Life), and mice were inoculated by the i.p. route with tachyzoites of each strain (in 200 μl volume) using a 28-gauge needle. Blood was collected by intracardiac puncture when the mice were euthanized. Animal euthanasia was completed in an approved $CO_2$ chamber. For immunolabeling on histological sections of the brains, the brains were removed from mice, entirely

embedded in a paraffin wax block and cut in 5-µm-thick layers using microtome. For statistical analysis of mouse survival data, the Mantel–Cox and Gehan–Breslow–Wilcoxon tests were used.

## Auxin-induced degradation

Depletion of MORC-mAID-HA was achieved with 3-indoleacetic acid (IAA, Sigma-Aldrich # 45533) used at 500 µM final concentration from a 500-mM stock solution prepared in EtOH, as described by Farhat et al, 2020 (Farhat et al, 2020). To monitor the degradation of AID-tagged proteins, parasites grown in HFF monolayers were treated with auxin for 24 to 48 h at 37 °C before parasites were harvested and analyzed by immunofluorescence or western blotting.

## HDAC3 inhibition using FR235222

FR235222 was provided by Astellas Pharma Inc. (Osaka, Japan) and dissolved into DMSO, and the final concentration in the culture medium was 50 ng/mL. Fifteen hours after infection of HFF monolayers, FR235222 was added to the medium and cells cultivated for 48 h.

## Immunofluorescence microscopy

*T. gondii*-infected HFF monolayers grown on coverslips were fixed in 3% formaldehyde for 20 min at room temperature, permeabilized with 0.1% (v/v) Triton X-100 for 15 min, and blocked in PBS containing 3% (w/v) bovine serum albumin (BSA). For immuno-labeling on histological sections of the brains, the brain layers spotted on glass slides were first solvent-dewaxed using toluene for three times 10 min and absolute alcohol for three times 10 min. The slides were then treated with citrate buffer pH 6, heated at 100 °C during 1 h, rinsed extensively with water and blocked in PBS containing 3% (v/v) BSA. The infected cells or brain layers were then incubated for 1 h with the primary antibodies indicated in the figures, followed by the addition of secondary antibodies conjugated to Alexa Fluor 488 or 594 (Molecular Probes) at a 1:1000 dilution for 1 h. The nuclei of both host cells and parasites were stained for 10 min at room temperature with Hoechst 33258 at 2 µg/mL in PBS. After four washes in PBS, coverslips were mounted on a glass slide with Mowiol mounting medium; images were acquired with a fluorescence ZEISS ApoTome.2 microscope and processed with the ZEN software (Zeiss).

## Western blot

Immunoblot analysis of protein was performed as follows: ~$10^7$ cells were lysed in 50 µl lysis buffer (10 mM Tris-HCl, pH 6.8, 0.5% SDS [v/v], 10% glycerol [v/v], and 1 mM EDTA and protease inhibitors cocktail) and sonicated. Proteins were separated by SDS-PAGE and transferred to a polyvinylidene fluoride membrane (PVDF, Immobilon-P; EMD Millipore) by liquid transfer 2 h at 110 V; blotted membranes were then probed using appropriate primary antibodies followed by alkaline phosphatase secondary antibodies (Life technologies). Band revelation was detected using NBT-BCIP (Thermo Fisher Scientific).

## Cyst observation

Fifty-nine days post-infection, the brain of each of the recipient mouse was homogenized in 2 mL of PBS. Images of cysts were

acquired between slide and slip cover with a fluorescence ZEISS ApoTome.2 microscope.

## Plaque assays

Confluent HFFs were infected with freshly egressed tachyzoites. Cultures were grown at 37 °C for 7 days, fixed, and stained with Coomassie blue staining solution (0.1% Coomassie R-250 in 40% ethanol and 10% acetic acid). The size of the plaques was measured using ZEN 2 Lite 9 software (Carl Zeiss, Inc.) and plotted using GraphPad Prism 8.

## Quantitative PCR

The parasite loads in the brain were quantified following DNA extraction (QiAmp DNA mini kit, Qiagen) using the quantitative PCR targeting of the *Toxoplasma*-specific 529-bp repeat element (Reischl et al, 2003). For statistical analysis of parasitic load differences between mice infected with 76K-GFP-luc and 76K-GFP-luc-Δ*bsm*, the nonparametric Wilcoxon-Mann-Whitney test was applied.

## Quantitative RT-PCR analysis of interleukins in the brain

Total RNA was isolated from brains using TRIzol (Thermo Fisher Scientific). First-strand cDNA was reverse transcribed from 50 to 100 ng small RNA using TaqMan microRNA reverse transcription kit (Applied Biosystems) and TaqMan probes (Applied Biosystems) for miR-155 (ID 002623), miR-146a (ID 000468). Quantitative PCR analyses of miRNAs were performed using Taqman miRNA expression (Applied Biosystems) assays according to the manufacturer's protocols in the ABI 7500 real-time PCR system (Applied Biosystems). Murine U6 snRNA/RNU6B (ID 001093) and RNU24 (ID 001001) were used as endogenous controls for normalization. Relative quantities of miRNA were analyzed by using the delta Ct method (Livak and Schmittgen, 2001).

## Mass spectrometry-based proteomic analyses

Proteins in the SDS-PAGE band between 62 and 98 kDa were in-gel digested with trypsin as previously described (Farhat et al, 2020). The resulting peptides were analyzed by online nanoliquid chromatography coupled to MS/MS (Ultimate 3000 RSLCnano and Q-Exactive Plus, Thermo Fisher Scientific) using a 120-min gradient. For this purpose, the peptides were sampled on a precolumn (300 µm × 5 mm PepMap C18, Thermo Scientific) and separated in a 75 µm × 250 mm C18 column (Reprosil-Pur 120 C18-AQ, 1.9 µm, Dr. Maisch). The MS and MS/MS data were acquired using Xcalibur 2.8 (Thermo Fisher Scientific). Peptides and proteins were identified by Mascot (version 2.8.3, Matrix Science) through concomitant searches against the *T. gondii* database (ME49 taxonomy, version 68 downloaded from ToxoDB), the Uniprot database (Homo sapiens taxonomy), and a homemade database containing the sequences of classical contaminant proteins found in proteomic analyses (human keratins, trypsin…). Trypsin/P was chosen as the enzyme and two missed cleavages were allowed. Precursor and fragment mass error tolerances were set at respectively at 10 and 20 ppm. Peptide modifications allowed during the search were: Carbamidomethyl (C, fixed), Acetyl (Protein N-term, variable), and Oxidation (M, variable). The

Proline software (version 2.2.0) was used for the compilation, grouping, and filtering of the results (conservation of rank 1 peptides, peptide length ≥6 amino acids, false discovery rate of peptide-spectrum-match identifications <1%, and minimum of one specific peptide per identified protein group). Proline was then used to perform a MS1 label-free quantification of the identified protein groups based on razor and specific peptides. Intensity-based absolute quantification (iBAQ) values were calculated for each protein group based on razor and specific peptides.

### *Toxoplasma gondii* transfection

*T. gondii* strains were electroporated with vectors in cytomix buffer (120 mM KCl, 0.15 mM CaCl$_2$, 10 mM K$_2$HPO$_4$/KH$_2$PO$_4$ pH 7.6, 25 mM HEPES pH 7.6, 2 mM EGTA, 5 mM MgCl$_2$) using a BTX ECM 630 machine (Harvard Apparatus). Electroporation was performed in a 2 mm cuvette at 1.100 V, 25 Ω and 25 µF. Drug selection was performed using pyrimethamine (3 µM). Single-clone of stable transgenic tachyzoites were obtained by limiting dilution in 96-well plates and verified by immunofluorescence assay or genomic analysis.

### Plasmid construction

The plasmids and primers for the genes of interest (GOI) used in this work are listed below. To construct the vector pLIC-GOI-HA-Flag, the coding sequence of GOI was amplified using primers LIC-GOI-Fwd and LIC-GOI-Rev using *T. gondii* genomic DNA as template. The resulting PCR product was cloned into the pLIC-HF-dhfr vector using the ligation-independent cloning (LIC) cloning method (Bougdour et al, 2013). For BSM disruption, the plasmid pTOXO_Cas9-CRISPR was described previously (Sangaré et al, 2016). Twenty mer-oligonucleotides corresponding BSM were cloned using the Golden Gate strategy. Briefly, primers BSM-gRNA-Fwd and BSM-gRNA-Rev containing the sgRNA targeting BSM genomic sequence were phosphorylated, annealed, and ligated into the pTOXO_Cas9-CRISPR plasmid linearized with BsaI, leading to pTOXO_Cas9-CRISPR::sgBSM.

### Gene synthesis for recombinant expression of *Tg*BSM

Gene synthesis for insect cell codon-optimized constructs was provided by Genscript. The original *T. gondii* *Tg*BSM construct (aa 1–763) was designed with a non-cleavable C-terminal 6His tag. The TgBCLA construct consisted of a 686 amino acid sequence designed to cover the N-terminal part followed by a conserved repetition, a degenerated repetition, and the C-terminal part of the BCLA protein encoded by TGME49_209755. The construct was also flanked by an N-terminal non-cleavable 6His tag. Both constructs were separately cloned between BamHI and HindIII sites into the pFastBac1 vector (Invitrogen).

### Generation of baculovirus

For both BSM and BCLA, bacmid cloning steps and baculovirus generation were performed using EMBacY baculovirus (kindly gifted by Imre Berger), which contains a YFP reporter gene in the virus backbone. The established standard cloning and transfection protocols setup within the EMBL Grenoble eukaryotic expression facility were used. Baculovirus synthesis (V0) and amplification (to

V1) were performed with SF21 cells cultured in ESF921 media (Expression system), large-scale expression cultures were performed with Hi-5 cells cultured in ESF921 media (Expression system) and infected with 0,5% vol/vol of generation 2 (V1) baculovirus suspensions and harvested 72 h post-infection.

### Protein expression and purification

**BSM**: For purification, the cell pellets of ~900 mL of Hi-5 culture were resuspended in 40 mL of lysis buffer (50 mM Tris pH 8.0, 500 mM NaCl and 4 mM β-mercapoethanol (β-ME)) in the presence of an anti-protease cocktail (Complete EDTA free, Roche) and 1 µl benzonase (MERK Millipore 70746). Lysis was performed on ice by sonication for 3 min (30 s on/ 30 s off, 45° amplitude). After the lysis step, 5% of glycerol was added. Clarification was then performed by centrifugation for 1 h at 12,000 × g and 4 °C. Twenty mM imidazole was then added to the supernatant and incubated with 3 mL of Ni-NTA resin (Qiagen) with a stirring magnet at 4 °C for 30 min. All further purification steps were then performed at room temperature. After flowing through the lysate, the resin was washed with 10 column volumes of lysis buffer containing 20 mM imidazole. Elution was then performed by increasing the imidazole content to 300 mM in a buffer system containing 200 mM NaCl, 50 mM Tris pH 7.5, 2 mM BME, and 5% glycerol. Eluted fractions were pooled based on an SDS-PAGE gel analysis and flown directly through a previously equilibrated (in 250 mM NaCl, 50 mM Tris pH 7.5, 2 mM BME and 5% glycerol) S200 column connected to an AKTA© pure system. Peak fractions were pooled and concentrated using a 50 kDa Amicon ultra (Sigma-Aldrich) concentrator. Glycerol concentration was adjusted to 20% before being frozen in liquid nitrogen and stored long-term at −80 °C.

**BCLA**: The protein expression, cell lysis, and Ni-NTA purification processes were identical for BCLA and BSM. After Ni-NTA purification, BCLA eluted fractions were pooled and dialyzed overnight in 50 mM NaCl, 50 mM Tris pH 7.5, 2 mM BME and 5% glycerol before injection on a Mono-Q column (GE Healthcare) preequilibrated with the same buffer as for dialysis. The column was eluted by a salt gradient (50 mM to 2 M NaCl) and peak fractions were pooled and injected on an S200 column as mentioned above for BSM.

### SEC-MALLS

The MALLS run was performed using an S200 Increase SEC column (10/300 GL, GE Healthcare). Sample injection and buffer flow were controlled by a Hitachi L2130 pump. The SEC column was followed by an L-2400 UV detector (Hitachi), an Optilab T-rEX refractometer (Wyatt technologies), and a DAWN HELEOS-II multi-angle light scattering detector (Wyatt technologies). Injections of 50 µL were performed using protein samples concentrated at a minimum of 4 mg.mL$^{-1}$, a constant flow rate of 0.5 mL.min$^{-1}$ was used. Accurate MALLS mass prediction was performed with the Astra software (Wyatt Technologies). The curve was plotted using Graphpad (Prism).

### Plate preparation

Midisorp plates (Nunc) were coated overnight (O.N) at 4 °C with either recombinant BSM or BCLA at 2 µg/mL in 100 mM calcium carbonate buffer pH 9.6 with 100 µl per well. After coating, plates

were washed twice with 250 µl of DPBS 0.05% Tween 20 (DPBS/Tween) then blocked for at least 2 h with 300 µl Superblock blocking buffer (Thermo Fisher) after which the buffer was removed and the plates dried upside down. Once dried, the plates could be stored for extended periods of time at 4 °C with no loss in serological reactivity.

## Sample preparation

All serum dilutions were prepared in DPBS 0.05% Tween 20, 0.1% BSA no more than 2 h prior to the assay. For mouse-tested sera, 1:400 dilutions were prepared. Eleven standards were also freshly prepared, consisting of 10 serial dilutions of a positive frozen stock serum set at 100 UI. Starting at a dilution 1:200 and following a ¾ dilution increment, the following titration points were prepared: 200 UI (1:200), 150 UI (1:266), 112.5 UI (1:356), 84.4 UI (1:474), 63.3 UI (1:632), 47.5 UI, (1:843) 35.6 UI (1:1124), 26.7 UI (1:1498), 20 UI (1:1998), and 15 UI (1:2663). A 0 UI standard was prepared with a dilution buffer only. For BCLA serology in humans, 1:400 sera dilutions were prepared. Six standards were also freshly prepared, consisting of five serial dilutions of an anti-His-tag chimeric human monoclonal antibody (Sigma-Aldrich, Saint Louis, USA). The following titration points were prepared: 200 UI, 100 UI, 50 UI, 25 UI, 12.5 UI. A 0 UI standard was prepared with a seronegative serum diluted at 1:400. For BSM serology in humans, sera were diluted at 1:200. Ten standards were also freshly prepared, consisting of nine serial dilutions of an anti-His-tag chimeric human monoclonal antibody (Sigma-Aldrich, Saint Louis, USA). The following titration points were prepared: 100 UI, 66.67 UI, 44.44 UI, 29.63 UI, 19.75 UI, 13.17 UI, 8.78 UI, 5.85 UI, 3.9 UI. A 0 UI standard was prepared with a dilution buffer only. The reactivity of a sample prepared with a dilution of anti-His-tag chimeric human monoclonal antibody (Sigma-Aldrich, Saint Louis, USA) at 10 UI was measured in each ELISA plate. BSM serology was calculated for each sample as the ratio between its own optical density (OD) and the OD of the cutoff from the same plate.

## ELISA assay

All the subsequent steps were implemented on the Gemini ELISA automation platform (Stratec). Dried plates were first washed twice with 350 µl of DPBS/Tween. Dilutions of the tested sera and standards were then distributed in the plates as row duplicates with 100 µl per well. Plates were then incubated 1 h at RT. After the incubation period, plates were washed 4 times with 350 µl of DPBS/Tween, 100 µl of peroxidase-coupled secondary antibody dilution (1:50,000 anti-mouse IgG or 1:50,000 anti-human IgG, Sigma-Aldrich ref A0168 and A0170, respectively) in DPBS 0.05% Tween 20, 0.1% BSA were then rapidly distributed in all wells. For human BSM serology, purified recombinant protein A/G peroxidase conjugated (Thermo Scientific) stored at −20 °C at 87 Units/mL was used diluted at 1:50,000 instead of anti-human IgG. After 1 h at RT, plates were washed four times in DPBS-Tween. Revelation reaction was performed by adding 100 µl of TMB Substrate (Thermo Fisher ref 34029) for 20 min precisely at RT then stopping the reaction with 50 µl of $H_2SO_4$ 0.2 M followed by 30 s of mixing. Well absorbance measurement was then performed using the Gemini integrated spectrophotometer at 450 nm.

## RNA-seq and sequence alignment

Total RNAs were extracted and purified using TRIzol (Invitrogen, Carlsbad, CA, USA) and RNeasy Plus Mini Kit (Qiagen). RNA quantity and quality were measured by NanoDrop 2000 (Thermo Scientific). For each condition, RNAs were prepared from three biological replicates. RNA integrity was assessed by standard non-denaturing 1.2% TBE agarose gel electrophoresis. RNA sequencing was performed following standard Illumina protocols, by Novogene (Cambridge, UK). Briefly, RNA quantity, integrity, and purity were determined using the Agilent 5400 Fragment Analyzer System (Agilent Technologies, Palo Alto, California, USA). The RQN ranged from 7.8 to 10 for all samples, which was considered sufficient. Messenger RNAs (mRNA) were purified from total RNA using poly-T oligo-attached magnetic beads. After fragmentation, the first-strand cDNA was synthesized using random hexamer primers. Then the second strand cDNA was synthesized using dUTP, instead of dTTP. The directional library was ready after end repair, A-tailing, adapter ligation, size selection, USER enzyme digestion, amplification, and purification. The library was checked with Qubit and real-time PCR for quantification and a bioanalyzer for size distribution detection. Quantified libraries will be pooled and sequenced on Illumina platforms, according to effective library concentration and data amount. The samples were sequenced on the Illumina NovaSeq platform (2 × 150 bp, strand-specific sequencing) and generated ~40 million paired-end reads for each sample. The quality of the raw sequencing reads was assessed using FastQC (www.bioinformatics.babraham.ac.uk/projects/fastqc/.babraham.ac.uk/projects/fastqc/) and MultiQC. For the expression data quantification and normalization, the FASTQ reads were aligned to the ToxoDB-49 build of the *T. gondii* ME49 genome using Subread version 2.0.1 with the following options 'subread-align -d 50 -D 600 --sortReadsByCoordinates'. Read counts for each gene were calculated using featureCounts from the Subread package. Differential expression analysis was conducted using DESeq2 and default settings within the iDEP.96 web interface. Transcripts were quantified and normalized using TPMCalculator. The Illumina RNA-seq dataset generated during this study is available at GSE271902.

## Nanopore direct RNA sequencing (DRS)

The mRNA library preparation followed the SQK-RNA002 kit (Oxford Nanopore)–recommended protocol, the only modification was the input mRNA quantity increased from 500 to 1000 ng, and all other consumables and parameters were standard. Final yields were evaluated using the Qubit HS dsDNA kit (Thermo Fisher Scientific, Q32851) with minimum RNA preps reaching at least 200 ng. For all conditions, sequencing was performed on FLO-MIN106 flow cells using a minION sequencer. All datasets were subsequently basecalled (high accuracy basecalling) with guppy v4.09 base-caller with a Q score cutoff of >7. Long read alignment was performed by Minimap2 as previously described (Antunes et al, 2024). Sam files were converted to bam and sorted using Samtools 1.4. Alignments were converted and sorted using Samtools 1.4.1. For the two samples, Toxoplasma aligned reads range between 300,000 and 500,000. The Nanopore DRS dataset is available at GSE271902.

## Data treatment

Blank subtractions were performed on duplicate blank wells for which primary antibody/sera were omitted but treated similarly as the others for all subsequent steps (washes, secondary Ab, substrate). Standard serum dilutions were averaged and fitted with a 4-parameter logistic regression with the upper asymptote value ($D_i$) fixed at 2.5 AU and all other variables ($A_i$, $B_i$, $C_i$) allowed to fit. For mouse and human BCLA serology and BSM mouse serology, tested dilution duplicates could have their apparent UI calculated and averaged from this regression; if in a duplicate measurement, the coefficient of variation was observed above 20%, then the sample would be re-tested. All the ELISA data presented in this work were obtained several times in independent titrations.

## Human sera

All experiments involving human samples were conducted in accordance with the ethical principles outlined in the World Medical Association (WMA) Declaration of Helsinki and the U.S. Department of Health and Human Services Belmont Report. This non-interventional, multicentric retrospective study involving data and samples from human participants was conducted at the Grenoble Alpes University Hospital using serum samples collected from both the Grenoble Alpes University Hospital and the Hospices Civils de Lyon, in accordance with French current regulation. The principal investigator (Dr Marie-Pierre Brenier-Pinchart, MD, PhD) has signed a commitment to comply with Reference Methodology n°MR004 issued by French Authorities (CNIL). Subjects were all informed and did not oppose; written consent for participation was not required for this study in accordance with the national legislation and the institutional requirements. The raw data supporting the conclusions of this article will be made available by the authors in respect of the General Data Protection Regulation, without undue reservation. All the sera were collected among sera received at the laboratory of the Grenoble Alpes University Hospital or the laboratory of the *Hospices Civils de Lyon* for toxoplasmosis serological routine analysis between December 1, 2011, and February 1, 2022, in agreement with the respective institutions and after patients' information and non-objection. Clinical data (age, sex, clinical and immune context) were collected and stored according to local ethic procedures. In Grenoble, analyses were performed using Vidas® Toxo IgM and IgG (bioMérieux, France) and Architect Toxo IgG and IgM (Abbott, Germany) in the Parasitology-Mycology Clinical Laboratory of the Grenoble Alpes University Hospital. In Lyon, analyses were conducted using Architect Toxo IgG and IgM (Abbott, Germany) and Vidas® Toxo IgG (bioMérieux, France) in the Parasitology-Mycology Clinical Laboratory of the Hospices Civils de Lyon. Briefly, the antibody titers for IgG were quantitatively expressed in IU/mL, whereas IgM were expressed as an index. The cutoffs defined by Vidas®, bioMérieux manufacturer were as follows: (i) IgG (IU/mL): negative < 4.0; 4.0 ≤ equivocal (gray zone) < 8; ≥ 8 positive; (ii) IgM (index): negative < 0.55; 0.55 ≤ equivocal (gray zone) < 0.65; ≥ 0.65 positive. The cutoffs defined by Architect®, Abbott manufacturer were as follows: (i) IgG (IU/mL): negative < 1.6; 1.6 ≤ equivocal (gray zone) < 3.0; ≥ 3.0 positive; (ii) IgM (index): negative < 0.50; 0.50 ≤ equivocal (gray zone) < 0.60; ≥ 0.60 positive. BCLA and BSM titers were measured on 357 selected patients' sera corresponding to two different serological status for toxoplasmosis: non-immunized

patients against toxoplasmosis (seronegative, $n = 134$ including 76 from Grenoble and 58 from Lyon) and patients with past immunity (chronic toxoplasmosis, $n = 222$ including 113 from Grenoble and 109 from Lyon). Among the patients with past immunity, those with a clinical context related primarily to reactivation from cysts were categorized into the subgroups "proven ocular toxoplasmosis" ($n = 23$) "cerebral toxoplasmosis" ($n = 5$) and "infraclinique reactivation" ($n = 6$). In addition, 115 sera from consecutive samples collected from 39 pregnant women who seroconverted during pregnancy and one man presenting with an acute infection were used to establish the kinetics of anti-bradyzoite serology. Of these precisely dated sera, 74 were collected within four months of infection and were assigned to the category "recent infection". Furthermore, BSM and BCLA ELISA serology titers was measured in sera collected at birth in children born to mothers who had seroconverted during pregnancy and for whom the diagnosis of congenital toxoplasmosis had been made ($n = 15$) or excluded ($n = 11$).

The absence of *T. gondii* immunity was concluded when the IgM- and IgG-specific antibody levels measured using Architect Toxo® IgG and Toxo® IgM assays were negative. Past immunity was considered when IgG were above the threshold of positivity with at least one method, and the IgM were negative or weakly positive (Architect® and Vidas®). Proved ocular toxoplasmosis (OT) was confirmed by detection of either *Toxoplasma* DNA using PCR and/or a local production of IgG and/or IgA antibodies by western blot (LDBIO Diagnostics, Lyon, France) (Greigert et al, 2019). Cerebral toxoplasmosis was diagnosed by PCR in cerebrospinal fluid in immunocompromised patients presenting acquired immunodeficiency syndrome (Brenier-Pinchart et al, 2022). These patients had clinical signs and radiological evidence of active disease. Infraclinical serological reactivations were observed in immunocompromised patients during the serological follow-up (Dard et al, 2018; Robert-Gangneux et al, 2018). In these patients, an increase of IgG levels compared to previous serological results were observed; furthermore, the *Toxoplasma*-PCR performed were negative, and these patients did not develop any clinical signs of toxoplasmosis (Fricker-Hidalgo et al, 2009). Seroconversion during pregnancy was objectified by the appearance of anti-*Toxoplasma* IgG or by a significant increase in IgG levels between two consecutive sera (Villard et al, 2016). Congenital toxoplasmosis in children has been confirmed by the presence of IgM, neosynthesized antibodies or persistence of IgG beyond 1 year.

## Statistics and reproducibility

Sample sizes were not predetermined and were chosen according to previous literature. Experiments were performed in biological replicates and provided consistent, statistically relevant results. No method of randomization was used. All experiments were performed in independent biological replicates as stated for each experiment in the manuscript. All corresponding treatment and control samples from ChIP-seq, RNA-seq, and ATAC-seq were processed at the same time to minimize technical variation. Investigators were not blinded during the experiments. Statistical significance was evaluated using $P$ values from unpaired two-tailed Student $t$ tests. Data are presented as the mean ± s.d. Significance was set to a $P$ value of <0.05. All the micrographs shown are representatives from three independently conducted experiments, with similar results obtained (Braun et al (2019); Gay et al (2016)).

**The paper explained**

**Problem**

*Toxoplasma gondii* is a brain-tropic parasite that causes toxoplasmosis, a food-borne disease. Although toxoplasmosis is often asymptomatic in the acute phase, it poses a serious risk to immunocompromised individuals and pregnant women. Congenital transmission can cause neurological damage, and reactivation in immunocompromised patients can be life-threatening. The long-term presence of the parasite in the brain has also been associated with psychiatric disorders. However, this chronic phase, which is typified by colonization of dormant cysts in the brain and muscles, is still poorly understood due to limited noninvasive detection methods.

**Results**

In this study, we introduce two new blood-based biomarkers that effectively identify mice carrying cysts and accurately differentiate between acute and chronic toxoplasmosis in humans. In addition, we reveal how chromatin-modifying proteins regulate specific transcription factors to support the development and long-term survival of *Toxoplasma* cysts in deep tissues, including the brain.

**Impact**

These findings improve serologic diagnosis of chronic toxoplasmosis and may aid in clinical decision-making, benefiting maternal–fetal health, immunocompromised patients, and neurological research.

## Graphics

Some of the figure and the synopsis graphics were created with BioRender.com.

## Data availability

The datasets produced in this study are available in the following databases: RNA-Seq data: Gene Expression Omnibus GSE287334. Chip-Seq data (Farhat et al, 2020): Gene Expression Omnibus GSE136060. ATAC-seq data: Gene Expression Omnibus GSE271901.

The source data of this paper are collected in the following database record: biostudies:S-SCDT-10_1038-S44321-025-00252-0.

## Peer review information

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

## Acknowledgements

We are grateful to the developers of the ToxoDB.org Genome Resource. ToxoDB and EuPathDB are part of the National Institutes of Health/National Institutes of Allergy and Infectious Diseases (NIH/NIAID)-funded Bioinformatics Resource Center. This work was supported by MSD Avenir [Project LatentToxoDiag, DS-2022-0017], the Laboratoire d'Excellence (LabEx) ParaFrap [ANR-11-LABX-0024], and the Agence Nationale pour la Recherche [Project ApiNewDrug, ANR-21-CE35-0010-01; Project ApiMORCing, ANR-21-CE15-0002-01; Project Chairs of Excellence in Biology/Health, Project ToxoNeoSex, ANR-24-CHBS-0008]. MS-based proteomic experiments were partially supported by Agence Nationale de la Recherche under projects ProFI (Proteomics French Infrastructure, ANR-10-INBS-08) and GRAL, a program from the Chemistry Biology Health (CBH) Graduate School of University Grenoble Alpes (ANR-17-EURE-0003). This work used the platforms of the Grenoble Instruct-ERIC center (ISBG; UAR 3518 CNRS-CEA-UGA-EMBL) within the Grenoble Partnership for Structural Biology (PSB), supported by FRISBI (ANR-10-INBS-0005-02) and GRAL, financed within the University Grenoble Alpes graduate school (Ecoles Universitaires de Recherche) CBH-EUR-GS (ANR-17-EURE-0003). We thank Caroline Mas for assistance and/or access to the biophysics platform.

## Author contributions

**Marie G Robert**: Conceptualization; Resources; Data curation; Formal analysis; Visualization; Methodology; Writing—original draft. **Christopher Swale**: Conceptualization; Data curation; Formal analysis; Investigation; Methodology. **Belen Pachano**: Formal analysis; Investigation; Visualization. **Léa Dépéry**: Resources; Formal analysis. **Valeria Bellini**: Resources; Formal analysis. **Céline Dard**: Resources; Formal analysis. **Dominique Cannella**: Resources; Formal analysis. **Charlotte Corrao**: Resources; Formal analysis. **Lucid Belmudes**: Resources; Formal analysis. **Yohann Couté**: Resources; Formal analysis; Supervision; Validation; Visualization. **Alexandre Bougdour**: Software; Visualization. **Hervé Pelloux**: Conceptualization; Supervision. **Emmanuelle Chapey**: Resources; Formal analysis. **Martine Wallon**: Conceptualization; Supervision. **Marie-Pierre Brenier-Pinchart**: Conceptualization; Data curation; Formal analysis; Supervision; Validation; Methodology; Writing—review and editing. **Mohamed-Ali Ha**kimi: Conceptualization; Formal analysis; Supervision; Funding acquisition; Validation; Investigation; Visualization; Methodology; Writing—original draft; Project administration; Writing—review and editing.

Source data underlying figure panels in this paper may have individual authorship assigned. Where available, figure panel/source data authorship is listed in the following database record: biostudies:S-SCDT-10_1038-S44321-025-00252-0.

## Disclosure and competing interests statement

MAH, CD, CS, HP, and MPBP are co-inventors of patent PCT/EP2020/081638, covering the use of BCLA as a biomarker for diagnosing chronic toxoplasmosis in humans. MAH, MGR, CS, and MPBP are also co-inventors of a related patent application on using BSM for the same purpose (patent EP24307283.2). Inserm and its subsidiary, Inserm-Transfert, are involved in the patent application. The remaining authors declare no competing interests.

# Expanded View Figures

**Figure EV1.  Identification of BSM in the MORC-depleted bradyzoite-enriched proteome.**

(**A**) Purification scheme. MORC-depleted extract was fractionated by chromatography as described in the Methods section. (**B**) Western blot analysis of the initial purification steps and S200 gel filtration fractions (F), using serum from an NMRI mouse chronically infected with the 76 K cystogenic strain of *Toxoplasma gondii*. (**C**) Histogram showing the expression levels of BSM (*TGME49_202020*) and *TGME49_216140* following MORC depletion or HDAC3 inhibition with FR235222. Transcript abundance is also displayed across various in vivo stages, including merozoites, EES1–EES5 stages, tachyzoites, sporozoites, and cysts. (**D**) The MORC KD strain was modified to express BSM or the protein encoded by TGME49_216140 with a C-terminal HA-Flag tag. Expression was assessed by IFA in untreated or IAA-treated parasites (24 h). Chimeric proteins were detected using FLAG staining (red). (**E, F**) FLAG immunoprecipitation eluates (**E**) of TGME49_216140 (**E**) or BSM (**F**) were probed with cyst-bearing mouse sera and/or FLAG antibodies.

▶

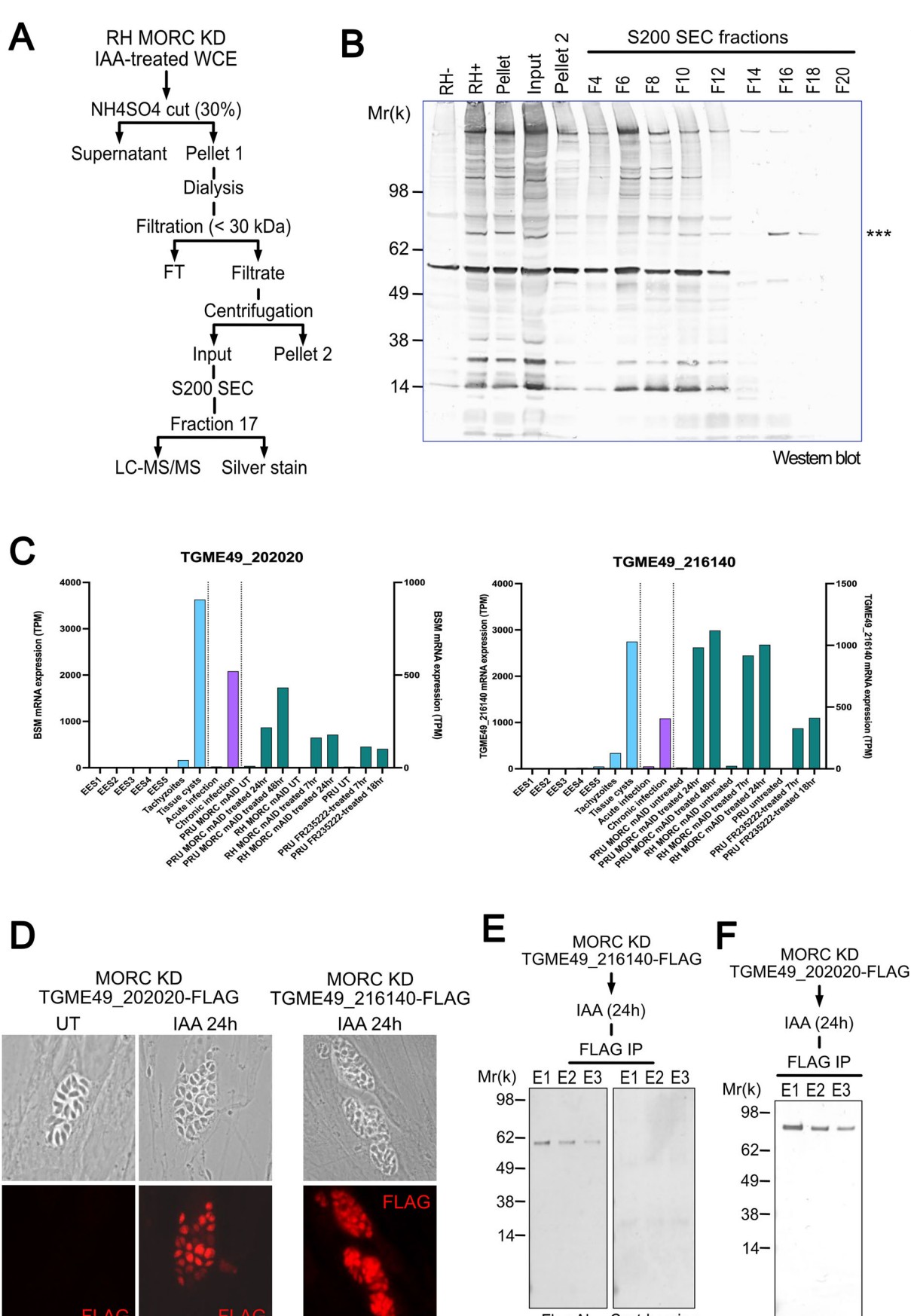

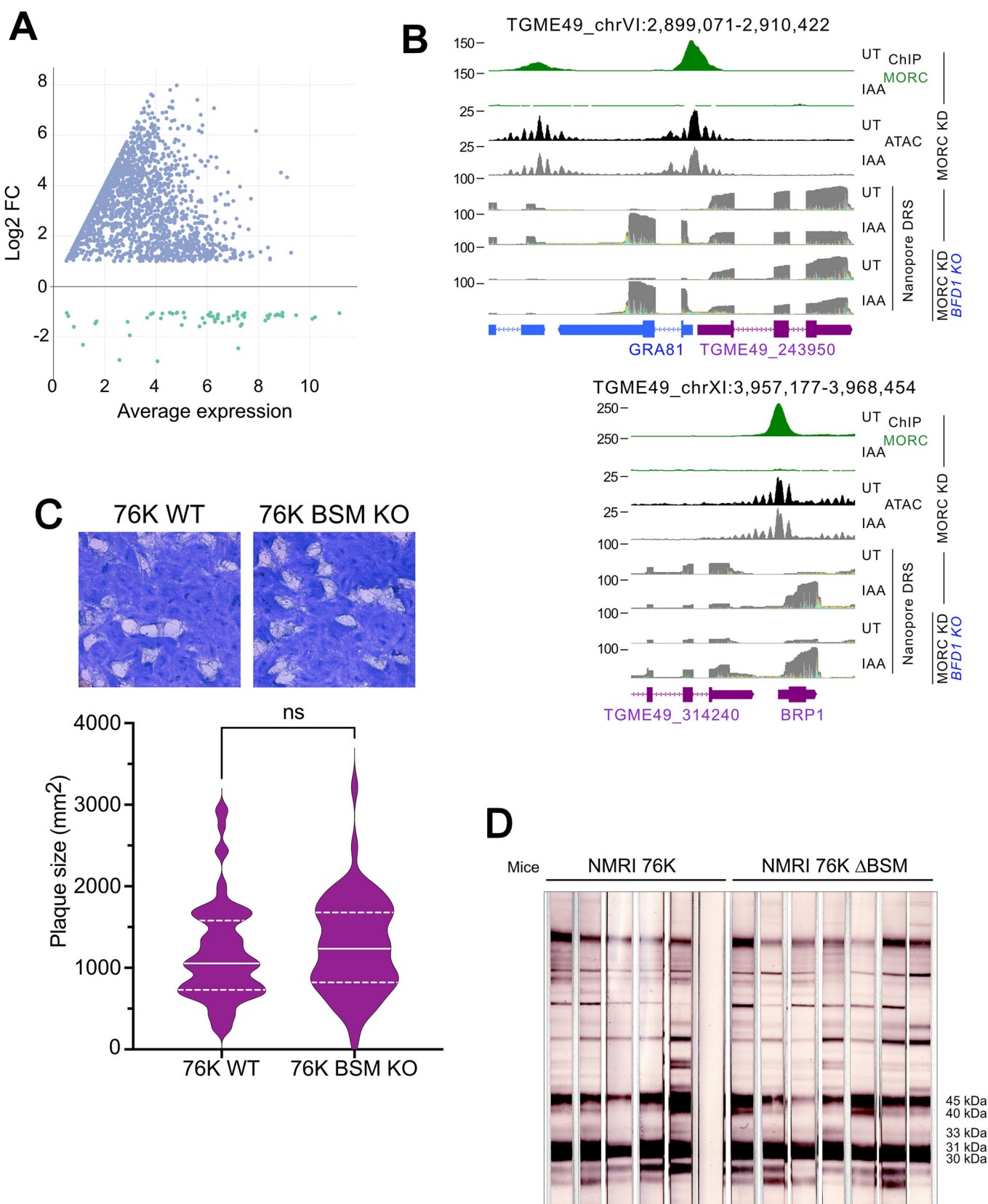

◄ **Figure EV2.   MORC regulome and BSM serology.**

(A) MA plots display Log2(FC) against Log2(mean expression) for genes before and after MORC depletion, using DESeq2. Upregulated genes (log2[FC] > 1, *P* value < 0.05) are blue, downregulated (log2[FC] < -0.58, *P* value < 0.05) are green. (B) IGB screenshots of representative genes expressed in merozoite (GRA81) or in merozoite/bradyzoite (BRP1) displaying ChIP-seq signal for MORC (HA antibody, green) in MORC KD strains under untreated and IAA-treated conditions. ATAC-seq profiles for both conditions, showing Tn5 transposase accessibility with read density on the y axis, are included. Nanopore DRS data for MORC KD and MORC KD/*BFD1* KO strains, untreated and IAA-treated, are also shown. (C) The effects of deletion of *BSM* on the lytic cycle were determined by plaque assay. After 7 days, the cells were fixed and stained with Coomassie blue to detect the presence of plaques (top panel). Graphs below show the distribution of the size of visible plaques (*n* = 50 per condition). Statistical analyses were performed using Mann–Whitney test. (D) LDBIO TOXO II IgG western blot membranes were probed with sera from NMRI mice 8 weeks post-infection with the 76 K wild-type strain (*n* = 6; two sera were insufficient) or ΔBSM (*n* = 7; one mouse died before 8 weeks). A test is considered positive if at least three of the 30, 31, 33, 40, and 45 kDa bands are present, including the 30 kDa band.

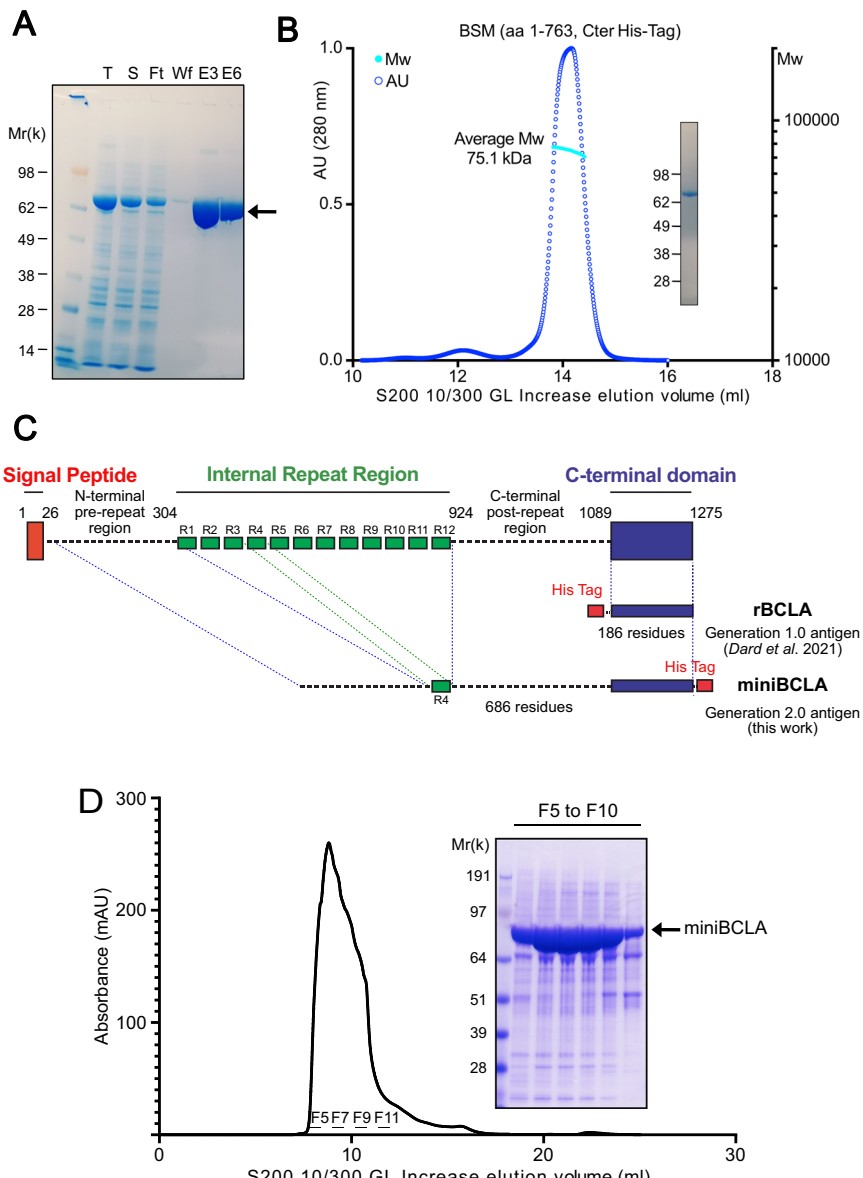

**Figure EV3. Recombinant BSM purification steps.**

(A) Nickel-nitrilotriacetic (Ni-NTA) elution. SDS-PAGE electrophoresis of total (T), soluble (S), flow though (Ft) and elution fractions 3 (E3) and 6 (E6). The black arrow points to the recombinant BSM protein. (B) Size Exclusion Chromatography using a S200 (10/300 Gl) combined to a Multi-Angle Laser Light Scattering analysis. Absorbance values are shown in deep blue while the predicted molecular weight plot (Mw) is displayed in light blue. (C) Schematic representation of rBCLA (1st generation of the antigen) and miniBCLA (this work) proteins. (D) Size exclusion chromatography of the purified miniBCLA antigen, 280 mm absorbance is shown as a function of volume. Peak elution fractions F5 to F10 were analyzed by Coomassie blue stained 4–12% NuPAGE.

