## [Peer Review File · EMBO Molecular Medicine]

Uncovering Biomarkers for Chronic Toxoplasmosis Detection Highlights Alternative Pathways Shaping Parasite Dormancy

Marie Robert, Christopher Swale, Belen Pachano, Lea Depery, Valeria Bellini, Céline DARD, Dominique Cannella, Charlotte Corrao, Lucid Belmudes, Yohann Couté, Alexandre Bougdour, Hervé Pelloux, Emmanuelle Chapey, Martine Wallon, Marie-Pierre Brenier-Pinchart, and Mohamed-ali HAKIMI

Corresponding authors: Mohamed-ali HAKIMI (mohamed-ali.hakimi@univ-grenoble-alpes.fr) , Marie-Pierre Brenier-Pinchart (MPPinchart@chu-grenoble.fr)

Review Timeline:

Submission Date:	4th Dec 24
Editorial Decision:	10th Jan 25
Revision Received:	22nd Mar 25
Editorial Decision:	12th Apr 25
Revision Received:	7th May 25
Accepted:	8th May 25

Editor: Zeljko Durdevic

Transaction Report:

10th Jan 2025

Dear Dr. Hakimi,

Thank you for the submission of your manuscript to EMBO Molecular Medicine. We have now received feedback from the three reviewers who agreed to evaluate your manuscript. All three referees recognize interest of the study but also raise important concerns that should be addressed in a major revision. If you would like to discuss further the points raised by the referees, I am available to do so via email or video. Let me know if you are interested in this option.

We would welcome the submission of a revised version within three months for further consideration. Please let us know if you require longer to complete the revision.

I look forward to receiving your revised manuscript.

Yours sincerely,

Zeljko Durdevic

We require:

- 1) A .docx formatted version of the manuscript text (including legends for main figures, EV figures and tables). Please make sure that the changes are highlighted to be clearly visible.
- 2) Individual production quality figure files as .eps, .tif, .jpg (one file per figure). For guidance, download the 'Figure Guide PDF': (<https://www.embopress.org/page/journal/17574684/authorguide#figureformat>).
- 3) A .docx formatted letter INCLUDING the reviewers' reports and your detailed point-by-point responses to their comments. As part of the EMBO Press transparent editorial process, the point-by-point response is part of the Review Process File (RPF), which will be published alongside your paper.
- 4) A complete author checklist, which you can download from our author guidelines (<https://www.embopress.org/page/journal/17574684/authorguide#submissionofrevisions>). Please insert information in the checklist that is also reflected in the manuscript. The completed author checklist will also be part of the RPF.
- 5) Please note that all corresponding authors are required to supply an ORCID ID for their name upon submission of a revised manuscript.
- 6) It is mandatory to include a 'Data Availability' section after the Materials and Methods. Before submitting your revision, primary datasets produced in this study need to be deposited in an appropriate public database, and the accession numbers and

database listed under 'Data Availability'. Please remember to provide a reviewer password if the datasets are not yet public (see <https://www.embopress.org/page/journal/17574684/authorguide#dataavailability>).

12) Author contributions: You will be asked to provide CRediT (Contributor Role Taxonomy) terms in the submission system. These replace a narrative author contribution section in the manuscript.

13) A Conflict of Interest statement should be provided in the main text.

14) Every published paper now includes a 'Synopsis' to further enhance discoverability. Synopses are displayed on the journal webpage and are freely accessible to all readers. They include a short stand first (maximum of 300 characters, including space) as well as 2-5 one-sentences bullet points that summarizes the paper. Please write the bullet points to summarize the key NEW findings. They should be designed to be complementary to the abstract - i.e. not repeat the same text. We encourage inclusion of key acronyms and quantitative information (maximum of 30 words / bullet point). Please use the passive voice. Please attach

these in a separate file or send them by email, we will incorporate them accordingly.

15) Include a Reagents and Tools Table as part of the Methods section, which can be downloaded from our author guidelines (<https://www.embopress.org/page/journal/17574684/authorguide#structuredmethods>)

***** Reviewer's comments *****

Referee #1 (Remarks for Author):

Summary

In the study by Robert, Swale et al., a new, reliable marker for the chronic stages of *Toxoplasma gondii* has been identified and analysed in great detail and shown to have a good potential for the development of diagnostic tests in the future.

The authors performed a tour de force in identifying the bradyzoite specific expression of BSM by taking advantage of a previously described knockdown for MORC generated by the same group. In this previous study the Hakimi group identified MORC to be critical for gene silencing and chromatin compaction, therefore acting as a key regulator throughout the development of the parasite. Since depletion of MORC results in the expression of bradyzoite-specific genes, the authors generated and analysed extracts from MORC depleted parasites, which led to the identification of BSM.

It is necessary to mention that a study on BSM (called DnaK-TPR (TGME49_202020) has been previously performed by the Bang group (Yang et al., 2017). In this study BSM has been shown to be specifically expressed in bradyzoites and that its deletion has no effect on cyst development (Yang et al., 2017). In this respect the current study can be seen as somewhat confirmatory. However, the Hakimi group goes far beyond the study of Yang et al., 2017, especially when it comes to the uncovering of the gene regulatory network regulating BSM and several other genes in a BFD1 independent manner. This work led to the suggestion that a new, hypothetical transcription factor is required for the regulation of a subset of bradyzoite-specific genes, that do not require the function of BFD1.

Indeed, the disruption of BFD1 specifically blocks MORC-driven induction of "only" a subset of bradyzoite-specific genes, while others are unaffected, clearly demonstrating that BFD1 does not act downstream of MORC/HDAC3 in the transcriptional regulation of many bradyzoite-specific genes, including BSM.

The diagnostic value of BSM and BCLA (which has been previously identified by the Hakimi group) is rigorously tested and compared. Briefly, an ELISA assay was developed and used to measure anti-BSM and anti-BCLA antibody titers in mice and shown that BSM serology is reliable and correlates with parasitic DNA in the brains of chronically infected mice and shows a very high accuracy.

Finally, the authors tested future application for human diagnostics, which led to mixed results that are relatively hard to interpret, due to relatively high variability of the results.

Own opinion

This is an excellent study and all the experiments and results, especially when it comes to the analysis of the mechanisms involved in regulation of BSM (and other bradyzoite-specific genes and their dependence on BFD1 and MORC) are crystal clear and the conclusions fully justified. This reviewer cannot find any flaws in this study and can only congratulate the authors to an excellent piece of work.

While there is a certain confirmatory aspect to the study from Yang et al., 2017, this reviewer believes that this study is well suited for EMBO Molecular Medicine.

My comments are very minor:

- Since TGME49_202020 has been previously called DnaK-TPR, the authors should consider using this name to avoid future confusion.
- All the figures are very well designed and of high quality. However, some panels are quite redundant and could be placed in the supplements in case space is an issue.

Referee #2 (Comments on Novelty/Model System for Author):

The technical quality is high. The novelty is high in that there is a new potential biomarker assessed and some new insights into regulation of bradyzoite development. However, the study does not focus on the regulatory function, and as such, the impact is probably limited/ of interest only to a specialised field.

While the markers may improve diagnosis of chronic Toxoplasma, it does not seem like a breakthrough for diagnostics. But I admit that I am not a specialist here and a medical doctor with expertise in Toxoplasma diagnosis would be better placed to judge the medical impact.

Referee #2 (Remarks for Author):

Uncovering Biomarkers for Chronic Toxoplasmosis Detection Highlights Alternative Pathways Shaping Parasite Dormancy
In this study the authors aim to identify novel bradyzoite markers for serological testing. To do this, they use a conditional mutant that they have previously generated to upregulate markers from "other-than-tachyzoite" stages, including bradyzoites. Using a series of elegant biochemical and genetic characterization, they identify 1 new marker that is expressed in bradyzoites (BSM), which they move on to test as a diagnostic marker for latent toxoplasma infection, comparing it to another marker that was recently described. In addition to the biomarker testing, they provide a series of experiments to test where in the hierarchy of events BSM falls when it comes to activation of cysts. They provide evidence that BSM is MORC dependent, but interestingly independent of the recently identified BFD1- which was thought to be one of the critical upstream regulators of bradyzoites and cyst formation.

Overall the approach is creative and the progress towards a better diagnostic marker important.

My major criticism is that the manuscript is hard to follow. Mixing of identification and development of BSM as a diagnostic marker and its regulation by MORC/ BFD1 is quite confusing and significantly distracts from the main aim. As such, I have rated the clarity and interest for the non-specialist as low. Of course, it is on the authors to decide to mix (in my humble view) 2 different stories, but if they want to keep it, they should better explain how the results influence each other. I could not really figure out why the section is relevant.

Figure 5: dBSM parasites forms normal cysts and the authors show that they have increased parasite load by qPCR. They state that these are probably latent stages because of increased miRNA expression levels. From these 2 points they conclude that BSM is probably involved in negatively regulating cysts burden in the host. I think the evidence is circumstantial- as the increased parasite load could also be Tachyzoites that re-emerge. Have the authors counted cysts between the mice infected with the different strains? I think they would need to do that for a solid statement. Otherwise they may want to take the statement out.

Minor:

Line 263: "BFD1 expression starts as early as 7 hours after FR235222 addition, peaks at 24 hours, and then declines at later time points (Fig. 2f), likely due to the simultaneous development of sexual stages (Antunes et al., 2023; Farhat et al., 2020)". The WB in figure 2f should have a loading control.

Line 288: The authors only show IFA images for 72h of Shield-1 treatment however say that BSM expression was triggered "regardless of the timing of treatment". Please add data for other treatment times/ or state which times were tested.

Line 295-296. In this sentence it is not clear that the authors compared the effect of MORC-KD in the presence and absence of a functional BFD1.

Line 296: correct "BFD1 Δ myb knockout" to "BFD1 Δ myb". Please correct Figure 1a: should it be MORC KD BFD1 Δ myb?

Figure 4b: There are other genes: TGME49_220560, TGME49_216140, TGME49_216140 that were also found to be upregulated by MORC depletion in the absence of a functional BFD1. Can the authors elaborate on this? Are these specifically expressed in the P6 cluster of Xue et al., 2020?

Line 314: "MORC was not detected at their promoters by ChIP-seq", "increased chromatin accessibility". Could the author provide metagene plots showing MORC-binding and Chromatin accessibility for the subset of genes exclusively expressed in chronic stages?

Part of the Figure 4.f is missing (gap in the green trace, second line from top)

Line 322 to 328: HDAC3 binding is not shown in Supplementary Figure. 2b and neither is the ROP26 gene (is GRA81 the same as ROP26?)

Referee #3 (Comments on Novelty/Model System for Author):

The findings have medium medical impact because the new antigen tests still don't have high specificity. This might not be a problem with the antigens though because there is no gold standard for comparison.

Referee #3 (Remarks for Author):

A substantial portion of the human population have been exposed to *T. gondii* based on serology, but determining which of these people are chronically infected with the parasite remains problematic because current serologic tests use antigens from the acute stage of the parasite. This well written manuscript reports the identification of a highly antigenic marker of *T. gondii* bradyzoites termed BSM, shows that recombinant BSM can be used to accurately identify chronically infected mice, and uses it to assess the serologic status of human serum samples from controls and infected people. A strength of the study is that it includes comparisons to a previously identified bradyzoite antigen BCLA along with improving the quality of such antigens for analysis. The study also provides compelling evidence that expression of different bradyzoite markers including BSM and BCLA are regulated distinctly. Overall, this study provides novel insights into stage specific gene expression and establishes the utility of using BSM and BCLA for improved serodiagnosis of *Toxoplasma* infection. These advances are important in the context of how little is known about chronic infection by a prevalent and medically important protozoan parasite. The study can be further improved by including description and discussion of prior studies that have used *T. gondii* MAG1 as a serologic antigen for chronic infection and by clarifying the nature of one of the transgenic strains used in the work.

Main

1. Although the Yolken lab has published several papers (PMC8678584, PMC4849725, PMC4005331) suggesting the utility of using MAG1 as an antigen for identifying chronically infected mice and people, the current study did not include a description of such studies in the introduction. This is necessary to convey a full picture of existing knowledge on the topic. It is also appropriate to discuss the current studies findings relative to those reported in the Yolkan lab papers.
2. It is unclear how BFD1 activity and that of associated proteins is affected in the MORC-KD-BFD1 Δ myb strain. This strain was made by replacing the BFD1myb domain with a DHFR selection cassette. Although it appears the intention was to create a strain wherein the DNA binding domain of BFD1 was disrupted, creating the strain by inserting a DHFR cassette could result in a variety of outcomes. The insertion likely resulted in an early stop codon in the BFD1 transcript, which could result either in expression of a truncated BFD1 protein or little or no expression of BFD1 if the truncated mRNA is degraded. From their RNAseq analysis the authors should be able to measure the extent to which expression of BFD1 mRNA is reduced. If there is little or no expression, then it would be more appropriate to describe this mutant as a knockdown or knockout rather than BFD1 Δ myb. Also, the authors should refrain from portraying this strain as being a "...disruption of the DNA-binding capacity of BFD1..." (line 279-280) since this is a misrepresentation of the strain. Alternatively, the authors could remake the strain with a more precise deletion of the BFD1 myb domain.

Minor

1. Line 88. *T. gondii* should be italicized.
2. Line 341. Please indicate why one mouse was excluded from the study.
3. Line 343. Consider using "infected" rather than "challenged" since the latter is often used to indicate infecting mice with preestablished immunity.
4. Line 569. Define QRS.

We sincerely thank you and the editorial board for the opportunity to resubmit a revised version of our manuscript. We are also grateful to the three reviewers for the time and effort they dedicated to evaluating our work. Their constructive feedback and insightful suggestions have been instrumental in improving the quality and clarity of our study. Please find below our detailed point-by-point responses, addressing each of the reviewers' comments. We have carefully revised the manuscript and incorporated additional data where necessary to strengthen our findings. We present our responses to each reviewer, addressing each point individually. Please excuse any repetition in our responses, as there were instances of overlapping comments or criticisms across the reviewers.

Referee #1 (Remarks for Author):

Summary

In the study by Robert, Swale et al., a new, reliable marker for the chronic stages of *Toxoplasma gondii* has been identified and analysed in great detail and shown to have a good potential for the development of diagnostic tests in the future. The authors performed a tour de force in identifying the bradyzoite specific expression of BSM by taking advantage of a previously described knockdown for MORC generated by the same group. In this previous study the Hakimi group identified MORC to be critical for gene silencing and chromatin compaction, therefore acting as a key regulator throughout the development of the parasite. Since depletion of MORC results in the expression of bradyzoite-specific genes, the authors generated and analysed extracts from MORC depleted parasites, which led to the identification of BSM. It is necessary to mention that a study on BSM (called DnaK-TPR (TGME49_202020) has been previously performed by the Bang group (Yang et al., 2017). In this study BSM has been shown to be specifically expressed in bradyzoites and that its deletion has no effect on cyst development (Yang et al., 2017). In this respect the current study can be seen as somewhat confirmatory. However, the Hakimi group goes far beyond the study of Yang et al., 2017, especially when it comes to the uncovering of the gene regulatory network regulating BSM and several other genes in a BFD1 independent manner. This work led to the suggestion that a new, hypothetical transcription factor is required for the regulation of a subset of bradyzoite-specific genes, that do not require the function of BFD1. Indeed, the disruption of BFD1 specifically blocks MORC-driven induction of "only" a subset of bradyzoite-specific genes, while others are unaffected, clearly demonstrating that BFD1 does not act downstream of MORC/HDAC3 in the transcriptional regulation of many bradyzoite-specific genes, including BSM. The diagnostic value of BSM and BCLA (which has been previously identified by the Hakimi group) is rigorously tested and compared. Briefly, an ELISA assay was developed and used to measure anti-BSM and anti-BCLA antibody titers in mice and shown that BSM serology is reliable and correlates with parasitic DNA in the brains of chronically infected mice and shows a very high accuracy. Finally, the authors tested future application for human diagnostics, which led to mixed results that are relatively hard to interpret, due to relatively high variability of the results.

Own opinion

This is an excellent study and all the experiments and results, especially when it comes to the analysis of the mechanisms involved in regulation of BSM (and other bradyzoite-specific genes and their dependence on BFD1 and MORC) are crystal clear and the conclusions fully justified. This reviewer cannot find any flaws in this study and can only congratulate the authors to an excellent piece of work. While there is a certain confirmatory aspect to the study from Yang et al., 2017, this reviewer believes that this study is well suited for EMBO Molecular Medicine.

- Thank you for your thoughtful and positive feedback. We appreciate your recognition of our work's clarity and rigor. We sincerely appreciate your support and are excited to contribute to EMBO Molecular Medicine.
- We had already referenced Yang et al., 2017, in our original manuscript in the following passage: “*This elevated parasite load may have been overlooked by Yang et al., who ended their experiment at 30 days instead of the two-month duration in our study (Yang et al., 2017). These findings suggest that BSM acts as a pro-host effector, helping to limit cyst burden.*” We have now also cited it in the introduction alongside the study by Ueno et al., 2011, which originally identified TGME49_202020 as DnaK-TPR.

My comments are very minor:

- Since TGME49_202020 has been previously called DnaK-TPR, the authors should consider using this name to avoid future confusion.

- TGME49_202020 was indeed previously identified as DnaK-TPR, named after its heat shock protein (DnaK) and tetratricopeptide repeat (TPR) domains (Ueno et al., 2011). Notably, other *Toxoplasma* proteins, including TGME49_294898, share a similar domain organization. While we acknowledge the prior naming of this protein, we believe that referring to it as BSM highlights its specificity to TGME49_202020 and more clearly reflects its role in the bradyzoite stage. That said, we have ensured that Ueno et al. (2011) and Yang et al. (2017) are appropriately credited in the introduction and throughout the main text.

- All the figures are very well designed and of high quality. However, some panels are quite redundant and could be placed in the supplements in case space is an issue.

- We thank the reviewer for the positive feedback. All panels remain within EMBO Molecular Medicine's figure limits, and supplementary figures have been renamed as EV Figures per journal guidelines.

Referee #2

(Comments on Novelty/Model System for Author): The technical quality is high. The novelty is high in that there is a new potential biomarker assessed and some new insights into regulation of bradyzoite development. However, the study does not focus on the regulatory function, and as such, the impact is probably limited/ of interest only to a specialized field. While the markers may improve diagnosis of chronic *Toxoplasma*, it does not seem like a breakthrough for diagnostics. But I admit that I am not a specialist here and a medical doctor with expertise in *Toxoplasma* diagnosis would be better placed to judge the medical impact.

Referee #2 (Remarks for Author):

Uncovering Biomarkers for Chronic Toxoplasmosis Detection Highlights Alternative Pathways Shaping Parasite Dormancy

In this study the authors aim to identify novel bradyzoite markers for serological testing. To do this, they use a conditional mutant that they have previously generated to upregulate markers from "other-than-tachyzoite" stages, including bradyzoites. Using a series of elegant biochemical and genetic characterization, they identify 1 new marker that is expressed in bradyzoites (BSM), which they move on to test as a diagnostic marker for latent toxoplasma infection, comparing it to another marker that was recently described. In addition to the

biomarker testing, they provide a series of experiments to test where in the hierarchy of events BSM falls when it comes to activation of cysts. They provide evidence that BSM is MORC dependent, but interestingly independent of the recently identified BFD1- which was thought to be one of the critical upstream regulators of bradyzoites and cyst formation. Overall, the approach is creative and the progress towards a better diagnostic marker important.

My major criticism is that the manuscript is hard to follow. Mixing of identification and development of BSM as a diagnostic marker and its regulation by MORC/ BFD1 is quite confusing and significantly distracts from the main aim. As such, I have rated the clarity and interest for the non-specialist as low. Of course, it is on the authors to decide to mix (in my humble view) 2 different stories, but if they want to keep it, they should better explain how the results influence each other. I could not really figure out why the section is relevant.

We thank the reviewer for this thoughtful comment. We acknowledge that the integration of BSM identification as a diagnostic marker with its regulation by MORC/BFD1 could initially appear as two separate narratives. However, we believe that these aspects are deeply interconnected, both mechanistically and clinically, and have made this rationale more explicit throughout the revised manuscript.

Our findings demonstrate that while BFD1 regulates bradyzoite genes such as BCLA, BSM is regulated independently, pointing to a multilayered and nuanced control of bradyzoite antigen expression. This complexity not only advances our understanding of parasite persistence but also strengthens the relevance of BSM and BCLA as complementary serologic markers, particularly for distinguishing recent versus past infections—an important challenge in the diagnosis of congenital toxoplasmosis and potentially in neuropsychiatric contexts.

We have thoroughly revised the manuscript to improve clarity and guide the reader through the interplay between molecular regulation and clinical relevance. This integrated narrative is now clearly reflected in key sections of the manuscript:

Revised abstract: “Mechanistic analyses show that the chromatin modifiers MORC and HDAC3 epistatically regulate BFD1, a key bradyzoite regulator. While BFD1 controls the expression of bradyzoite genes such as BCLA, a specific subset, including BSM, is regulated independently of BFD1. This multilayered regulation complicates the understanding of parasite persistence in humans, but offers promise for improved serologic diagnosis during pregnancy, but also in individuals with mental illness.”

Introduction: “This finding underscores the diversity of molecular pathways and their regulatory checkpoints that control cyst development and persistence in tissues, leading to a range of immune responses in hosts and complicating diagnosis of chronic toxoplasmosis in humans.”

Discussion: “While MORC/HDAC3 restricts gene expression through chromatin accessibility, BFD1 selectively regulates bradyzoite-specific genes, excluding BSM, which appears independently controlled. This regulatory distinction supports a nuanced view of bradyzoite immunogenicity, where BSM and BCLA play different roles in immune recognition. This underscores the need for a broad, unbiased exploration of bradyzoite proteins to uncover potential serologic markers.”

Altogether, we hope this revised version better conveys the scientific coherence and relevance of combining mechanistic and diagnostic insights in a unified manuscript.

Figure 5: dBSM parasites forms normal cysts and the authors show that they have increased parasite load by qPCR. They state that these are probably latent stages because of increased miRNA expression levels. From these 2 points they conclude that BSM is probably involved in negatively regulating cysts burden in the host. I think the evidence is

circumstantial- as the increased parasite load could also be Tachyzoites that re-emerge. Have the authors counted cysts between the mice infected with the different strains? I think they would need to do that for a solid statement. Otherwise, they may want to take the statement out.

To avoid overinterpretation, we have revised the corresponding paragraph in the manuscript to attenuate our conclusion. We now present the increased parasite load observed in Δ BSM-infected brains as a possible indication that BSM may modulate parasite persistence, while explicitly acknowledging that further analyses, including cyst quantification, would be required to clarify the nature of this effect.

Main text: "This increased parasite burden, which may have been overlooked by Yang et al. due to the shorter time period in their study (30 days versus two months in our study; Yang et al., 2017), raises the possibility that BSM may play a role in modulating parasite persistence. However, additional analyses, including direct quantification of cysts, would be required to clarify the nature of this increased parasite burden and whether it reflects a difference in cyst load or potential parasite reactivation."

Minor:

Line 263: "BFD1 expression starts as early as 7 hours after FR235222 addition, peaks at 24 hours, and then declines at later time points (Fig. 2f), likely due to the simultaneous development of sexual stages (Antunes et al., 2023; Farhat et al., 2020)". The WB in figure 2f should have a loading control.

- We have added GAP45 as a parasite loading control in a revised Fig. 2f.

Line 288: The authors only show IFA images for 72h of Shield-1 treatment however say that BSM expression was triggered "regardless of the timing of treatment". Please add data for other treatment times/ or state which times were tested.

- We agree that the original wording was unclear. In our experiments, we used 72 hours of Shield-1 treatment, which corresponds to the optimal time point for robust BFD1 expression and maturation of bradyzoite cysts capable of expressing bradyzoite markers. This time point was determined based on our own optimization and is also supported by previous studies from Sebastian Lourido's group. Accordingly, we used this standardized 72-hour induction time across all experiments and have now revised the manuscript to replace the phrase "regardless of the timing of treatment" with a precise mention of the 72-hour induction time.

Line 295-296. In this sentence it is not clear that the authors compared the effect of MORC-KD in the presence and absence of a functional BFD1.

- We agree that the original sentence was unclear and have now revised the text for clarity. The sentence now reads: "(...) sequencing of bulk mRNA in both MORC KD and MORC KD BFD1 KO strains." to explicitly indicate that we compared the effect of MORC depletion in the presence and absence of BFD1.

Line 296: correct "BFD1 Δ myb knockout" to "BFD1 Δ myb".

- In response to Reviewer 3 (see below), we have revised the text and all relevant figures to consistently refer to this strain as MORC KD/BFD1 KO, instead of BFD1 Δ myb.

Please correct Figure 1a: should it be MORC KD BFD1Δmyb?

- There is no error in Fig. 1a. As indicated in the panel, the purification was performed from extracts of the MORC KD (mAID-HA) strain.

Figure 4b: There are other genes: TGME49_220560, TGME49_216140, TGME49_216140 that were also found to be upregulated by MORC depletion in the absence of a functional BFD1. Can the authors elaborate on this? Are these specifically expressed in the P6 cluster of Xue et al., 2020?

As some genes show relatively low expression levels, it can be challenging to confidently assess their distribution in single-cell datasets—for example, TGME49_220560 appears weakly expressed, whereas TGME49_216140 shows much stronger expression (See Figure below). Nevertheless, for TGME49_216140, we confirm both its presence in the P6 cluster and its BFD1-independent regulation upon MORC depletion. These findings suggest a broader trend, but we prefer to remain cautious in generalizing this without performing dedicated single-cell experiments—work that is currently underway in our lab.

TGME49_220560

TGME49_216140

Line 314: "MORC was not detected at their promoters by CHIP-seq", "increased chromatin accessibility". Could the author provide metagene plots showing MORC-binding and Chromatin accessibility for the subset of genes exclusively expressed in chronic stages?

In response, we have added metagene plots for MORC binding (revised Fig. 4d) and chromatin accessibility (ATAC-seq) upon MORC depletion (revised Fig. 4e), comparing genes specifically expressed in the bradyzoite subset to those from the merozoite subset. Our initial conclusions remain unchanged: MORC is not detected near bradyzoite genes, yet its depletion leads to increased chromatin accessibility, suggesting the involvement of secondary transcription factors regulated upstream by MORC. This hypothesis is further supported by our previous studies (Farhat et al., 2020; Antunes et al., 2024).

Part of the Figure 4.f is missing (gap in the green trace, second line from top)

Thank you for your observation. The data in Figure 4.f are visualized using the Integrative Genomics Browser (IGB). The apparent gap in the green trace likely reflects the outcome of background subtraction or signal normalization applied during data processing. In some cases, peak calling software or signal track generation tools (such as MACS2 or deepTools) can filter out regions with low or negligible enrichment, resulting in a flat or absent signal trace—even if some raw reads are still present. This typically occurs in genomic regions where ChIP enrichment is weak or absent, and thus is consistent with the biological signal.

Line 322 to 328: HDAC3 binding is not shown in Supplementary Figure. 2b and neither is the ROP26 gene (is GRA81 the same as ROP26?)

In the text (lines 322–328), we refer to data showing local co-occupancy of MORC and HDAC3 at genes expressed in both merozoites and bradyzoites, such as BRP1 and ROP26, illustrating a canonical 'poised' bivalent chromatin state (Supplementary Figure. 2b now revised Fig. EV2B). While the figure provides a representative example, additional evidence for HDAC3 binding and poised chromatin architecture at these loci, including ROP26, is presented in our previous studies (Antunes et al., 2024; Farhat et al., 2020). We encourage readers to refer to these references for a more detailed view of HDAC3 recruitment following MORC binding near BRP1, ROP26, and other merozoite-specific genes.

(And to clarify: *GRA81* and *ROP26* are distinct genes.)

Referee #3

(Comments on Novelty/Model System for Author): The findings have medium medical impact because the new antigen tests still don't have high specificity. This might not be a problem with the antigens though because there is no gold standard for comparison.

(Remarks for Author):

A substantial portion of the human population have been exposed to *T. gondii* based on serology, but determining which of these people are chronically infected with the parasite remains problematic because current serologic tests use antigens from the acute stage of the parasite. This well written manuscript reports the identification of a highly antigenic marker of *T. gondii* bradyzoites termed BSM, shows that recombinant BSM can be used to accurately identify chronically infected mice, and uses it to assess the serologic status of human serum samples from controls and infected people. A strength of the study is that it includes comparisons to a previously identified bradyzoite antigen BCLA along with improving the quality of such antigens for analysis. The study also provides compelling evidence that expression of different bradyzoite markers including BSM and BCLA are regulated distinctly. Overall, this study provides novel insights into stage specific gene expression and establishes the utility of using BSM and BCLA for improved serodiagnosis of *Toxoplasma* infection. These advances are important in the context of how little is known about chronic infection by a prevalent and medically important protozoan parasite. The study can be further improved by including description and discussion of prior studies that have used *T. gondii* MAG1 as a serologic antigen for chronic infection and by clarifying the nature of one of the transgenic strains used in the work.

Main

1. Although the Yolken lab has published several papers (PMC8678584, PMC4849725, PMC4005331) suggesting the utility of using MAG1 as an antigen for identifying chronically infected mice and people, the current study did not include a description of such studies in the introduction. This is necessary to convey a full picture of existing knowledge on the topic. It is also appropriate to discuss the current study's findings relative to those reported in the Yolken lab papers.

We sincerely apologize for having overlooked the inclusion of these studies in our introduction and have now addressed this oversight.

The first study linking Yolken to the use of MAG1 peptides for detecting chronicity in a murine model was published in 2013 (PMC4005331), well before subsequent research demonstrated that MAG1 is expressed in both tachyzoites and bradyzoites. However, contrary to the assumption that MAG1 is a specific marker of the bradyzoite/cyst stage, strong evidence indicates its broad expression across both acute and chronic *Toxoplasma* stages.

Ferguson et al. reported the presence of MAG1 protein in tachyzoites, although with a lower expression level with respect to cysts (Ferguson and Parmley, Trends Parasitol. 2002).

In line with this observation and as illustrated in Fig. 1, MAG1 mRNA is strongly expressed in tachyzoites (>2000 TPM), though its levels are indeed higher in bradyzoites/cysts (>5000 TPM) (Fig. 1a). This pattern contradicts the idea that MAG1 is exclusive to the chronic stage. Additionally, MAG1 is clearly detected as a protein in dense granules isolated from tachyzoites using hyperLOPIT (Fig. 1b), further reinforcing its expression beyond the bradyzoite/cyst stage. In contrast, neither BCLA nor BSM were detected by hyperLOPIT in tachyzoites.

Figure 1. (a)

(b)

These findings collectively undermine the idea that MAG1 serves as a reliable marker for chronic infection, as its expression in tachyzoites contradicts its classification as a bradyzoite-specific protein. Yet, we do not dispute the possibility that MAG1 could be detected serologically with greater intensity during chronic toxoplasmosis.

We appreciate the reviewer's insightful comment. We have expanded the introduction to provide a more comprehensive overview of previous studies on this topic, placing our findings in the broader context of existing literature. As suggested, we have also incorporated all relevant contributions, including those in PMC8678584, PMC4849725, and PMC4005331.

2. It is unclear how BFD1 activity and that of associated proteins is affected in the MORC-KD-BFD1 Δ myb strain. This strain was made by replacing the BFD1myb domain with a DHFR selection cassette. Although it appears the intention was to create a strain wherein the DNA binding domain of BFD1 was disrupted, creating the strain by inserting a DHFR cassette could result in a variety of outcomes. The insertion likely resulted in an early stop codon in the BFD1 transcript, which could result either in expression of a truncated BFD1 protein or little or no expression of BFD1 if the truncated mRNA is degraded. From their RNAseq analysis the authors should be able to measure the extent to which expression of BFD1 mRNA is reduced. If there is little or no expression, then it would be more appropriate to describe this mutant as a knockdown or knockout rather than BFD1 Δ myb. Also, the authors should refrain from portraying this strain as being a "...disruption of the DNA-binding capacity

of BFD1..." (line 279-280) since this is a misrepresentation of the strain. Alternatively, the authors could remake the strain with a more precise deletion of the BFD1 myb domain.

We appreciate the reviewer's insightful comment. We acknowledge that replacing the coding sequence of the BFD1 Myb domain with a DHFR selection cassette effectively results in a BFD1 knockout (KO) rather than a precise deletion limited to the Myb domain. Consequently, this strain does not solely disrupt the DNA-binding capacity of BFD1 but instead leads to a complete loss of BFD1 expression or the potential production of a truncated, non-functional protein, depending on mRNA stability.

To address this concern, we have reanalyzed our RNA-seq data to assess BFD1 transcript levels in this strain. Our analysis confirms that BFD1 expression is significantly reduced (or absent), supporting the notion that this strain functions as a BFD1 KO rather than a targeted Myb domain deletion. Based on this, we have revised the text and all relevant figures to accurately describe this strain as MORC-KD BFD1-KO rather than BFD1 Δ myb.

Furthermore, we have corrected the misleading statement regarding the disruption of BFD1's DNA-binding capacity (lines 279–280) to ensure it properly reflects the nature of the genetic modification.

Minor

1. Line 88. *T. gondii* should be italicized.

The revised manuscript has been updated accordingly.

2. Line 341. Please indicate why one mouse was excluded from the study.

The revised manuscript now clarifies the reason for the exclusion of one mouse. The sentence has been updated to read: "All mice showed a strong adaptive immune response, as shown by Western blots (Fig. EV2D), except one that was likely poorly infected during parasite inoculation and was therefore excluded from the study."

3. Line 343. Consider using "infected" rather than "challenged" since the latter is often used to indicate infecting mice with preestablished immunity.

The revised manuscript has been updated accordingly.

4. Line 569. Define QRS.

QRS stands for Glutaminyl-tRNA synthetase, as outlined in the revised manuscript.

12th Apr 2025

Dear Dr. Hakimi,

Thank you for the submission of your revised manuscript to EMBO Molecular Medicine. I am pleased to inform you that we will be able to accept your manuscript pending the following final amendments:

- 1) Figures: Please remove panel Figure 1A from the manuscript as previously discussed with our Data Integrity Analyst, Christopher Rickerby.
 - 2) In the main manuscript file, please do the following:
 - Please address all comments suggested by our data editors listed below:
 - o Figure legends:
 1. Please note that the exact p values are not provided in the legends of figures 5F, 6A, B.
 2. Please indicate the statistical test used for data analysis in the legend of figure 6E.
 3. Please note that information related to n is missing in the legends of figures 5C-F; EV2 C.
 4. Please note that the error bars are not defined in the legends of figures 5C-F.
 5. Please note that the scale bar needs to be defined for figures 2A, B; 3B, C.
 6. Please note that the black arrows are not defined in the legend of figure EV3 A. This needs to be rectified.
 - Add callouts for Figure 4G.
 - Author contributions: Please remove it from the manuscript and specify author contributions in our submission system. CRediT has replaced the traditional author contributions section because it offers a systematic machine-readable author contributions format that allows for more effective research assessment. You are encouraged to use the free text boxes beneath each contributing author's name to add specific details on the author's contribution. More information is available in our guide to authors:
<https://www.embopress.org/page/journal/17574684/authorguide#authorshipguidelines>
 - Indicate in legends exact n and exact p values, not a range, along with the statistical test used. To keep the figures "clear" some authors found providing an Appendix table Sx with all exact p-values preferable. You are welcome to do this if you want to.
 - In Methods, provide the antibody dilutions that were used for each antibody.
 - In Methods, add the following paragraph:
- Graphics:
- (some of the... OR Figure #... OR synopsis) Graphics were created with BioRender.com.
- In Methods, please include statement that the experiments with human samples conformed to the principles set out in the WMA Declaration of Helsinki and the Department of Health and Human Services Belmont Report.
 - Please remove reagents and tools table from Methods and leave only the uploaded file.
 - Please remove the sentence "Correspondence and requests for materials should be addressed to M.A.H." from data availability statement.
- 3) Movie: Rename it to Movie EV1 and upload the file zipped together with its legend. Please update its callouts in the main text.
 - 4) Datasets: Rename datasets to Dataset EV1-EV4 and add their legends in a separate tab. Also, update their callouts in the main text.
 - 5) Funding: Please make sure that information about all sources of funding are complete in both our submission system and in the manuscript. Currently, ANR-17-EURE-0003, UAR 3518 CNRS-CEA-UGA-EMBL, ANR-10-INBS-0005-02 and ANR-17-EURE-0003 are missing in our system. Please correct.
 - 6) Synopsis:
 - Synopsis image: Please resize the image to 550 px-wide x (300-600)-px high and upload it as a high-resolution jpeg file.
 - Please check your synopsis text and image before submission with your revised manuscript. Please be aware that in the proof stage minor corrections only are allowed (e.g., typos).
 - 7) Source data: Please upload source data as one zipped folder per figure.
 - 8) As part of the EMBO Publications transparent editorial process initiative (see our Editorial at <http://embomolmed.embopress.org/content/2/9/329>), EMBO Molecular Medicine will publish online a Review Process File (RPF) to accompany accepted manuscripts. This file will be published in conjunction with your paper and will include the anonymous referee reports, your point-by-point response and all pertinent correspondence relating to the manuscript. Let us know whether you agree with the publication of the RPF and as here, if you want to remove or not any figures from it prior to publication. Please note that the Authors checklist will be published at the end of the RPF.
 - 9) Please provide a point-by-point letter INCLUDING my comments as well as the reviewer's reports and your detailed responses (as Word file).

I look forward to reading a new revised version of your manuscript as soon as possible.

Yours sincerely,

Zeljko Durdevic

*** Instructions to submit your revised manuscript ***

- 1) a .docx formatted version of the manuscript text (including Figure legends and tables)
- 2) Separate figure files*
- 3) supplemental information as Expanded View and/or Appendix. Please carefully check the authors guidelines for formatting Expanded view and Appendix figures and tables at <https://www.embopress.org/page/journal/17574684/authorguide#expandedview>
- 4) a letter INCLUDING the reviewer's reports and your detailed responses to their comments (as Word file).
- 5) The paper explained: EMBO Molecular Medicine articles are accompanied by a summary of the articles to emphasize the major findings in the paper and their medical implications for the non-specialist reader. Please provide a draft summary of your article highlighting
 - the medical issue you are addressing,
 - the results obtained and
 - their clinical impact.This may be edited to ensure that readers understand the significance and context of the research. Please refer to any of our published articles for an example.
- 6) Author contributions: the contribution of every author must be detailed in a separate section.
- 7) EMBO Molecular Medicine now requires a complete author checklist (<https://www.embopress.org/page/journal/17574684/authorguide>) to be submitted with all revised manuscripts. Please use the checklist as guideline for the sort of information we need WITHIN the manuscript. The checklist should only be filled with page numbers where the information can be found. This is particularly important for animal reporting, antibody dilutions (missing) and exact values and n that should be indicated instead of a range.
- 8) Every published paper now includes a 'Synopsis' to further enhance discoverability. Synopses are displayed on the journal webpage and are freely accessible to all readers. They include a short stand first (maximum of 300 characters, including space) as well as 2-5 one sentence bullet points that summarise the paper. Please write the bullet points to summarise the key NEW findings. They should be designed to be complementary to the abstract - i.e. not repeat the same text. We encourage inclusion of key acronyms and quantitative information (maximum of 30 words / bullet point). Please use the passive voice. Please attach these in a separate file or send them by email, we will incorporate them accordingly.

You are also welcome to suggest a striking image or visual abstract to illustrate your article. If you do please provide a jpeg file 550 px-wide x 300-600px high.

9) A Conflict of Interest statement should be provided in the main text

10) Please note that we now mandate that all corresponding authors list an ORCID digital identifier. This takes <90 seconds to complete. We encourage all authors to supply an ORCID identifier, which will be linked to their name for unambiguous name identification.

Currently, our records indicate that the ORCID for your account is 0000-0002-2547-8233.

Link Not Available

11) Include a Reagents and Tools Table as part of the Methods section, which can be downloaded from our author guidelines (<https://www.embopress.org/page/journal/17574684/authorguide#structuredmethods>)

Photos 400-800 DPI

*Additional important information regarding figures and illustrations can be found at

<https://bit.ly/EMBOPressFigurePreparationGuideline>. See also figure legend preparation guidelines:

<https://www.embopress.org/page/journal/17574684/authorguide#figureformat>

***** Reviewer's comments *****

Referee #2 (Comments on Novelty/Model System for Author):

The manuscript has significantly increased in clarity. It is now suitable (in my view), for publication.

Referee #2 (Remarks for Author):

The manuscript has significantly increased in clarity. It is now suitable (in my view), for publication. Congratulations on the work

Referee #3 (Comments on Novelty/Model System for Author):

This is a very thorough study that yields novel insights into Tg chronic infection with relevance to improving the diagnosis of chronic infection.

Referee #3 (Remarks for Author):

The authors have thoroughly addressed the comments of this reviewer and are commended on a lovely piece of work.

The authors addressed the remaining editorial issues.

8th May 2025

Dear Dr. Hakimi,

We are pleased to inform you that your manuscript is accepted for publication and is now being sent to our publisher to be included in the next available issue of EMBO Molecular Medicine.

Zeljko Durdevic
Senior Editor
EMBO Molecular Medicine
